# Lottery before peer review is associated with increased female representation and reduced estimated economic cost in a German funding line

Finn Luebber [1,2], Sören Krach [1,2] ✉, Frieder M. Paulus [1,2], Lena Rademacher [1,2,3] & Rima-Maria Rahal [4,5]

Research funding is a key determinant of scientific progress. However, current allocation procedures for third-party funding are criticized due to high costs and biases in the selection. Here, we present data from a large German funding organization on an implementation of a lottery-first approach followed by peer review to allocate funding. We examine the changes in submissions and funded projects of female applicants after implementation, estimate the costs of the overall allocation process, and report on the attitudes and satisfaction of researchers and reviewers. The data show an increase of 10% in submissions and a 23% increase in funded projects from female applicants with the lottery-first approach compared to a previously used procedure. Additionally, the lottery-first approach was estimated to have 68% lower economic costs compared to a conventional single-stage peer review approach. Satisfaction with this funding approach was high and around half of applicants preferred an initial lottery followed by peer review over a conventional approach. Thus, the lottery-first approach is a promising addition to allocation procedures.

The current approach to determining which research and higher education projects receive funding has serious drawbacks, particularly in terms of costs and bias[1-4]. These flawed (i.e., biased and inefficient) procedures are especially problematic given the substantial investments societies make in research: on average, OECD countries allocate 2.7% of their gross domestic product to scientific research, totaling ~$1.7 trillion annually in research investments. Competitive third-party funding, i.e., funding from public (state, federal, international) or private sources, in addition to basic researcher and institutional funding, comprises a large portion of total research investment. For example, competitive third-party funding accounted for 28% of the total German research budget in 2022, following a marked increase. As reliance on competitive third-party funding grows[5], there have been increasing calls for more streamlined and less biased funding allocation processes to reduce the large opportunity costs in lost research output[6,7].

Traditionally, competitive third-party funding schemes rely on reviewer panels to determine which projects to fund following at least one round of peer review—the 'gold standard' for funding allocation in research and higher education[8]. To submit a project for consideration, researchers typically prepare a proposal outlining the project, which is then reviewed by multiple peers. This conventional approach results in a significant portion of academic working hours being devoted to writing and evaluating project proposals[9-11]. As a result, it is costly for applicants, institutions, and society at large: applicants cannot use the

[1]Department of Psychiatry and Psychotherapy, University of Lübeck, Lübeck, Germany. [2]Open Science Initiative, University of Lübeck, Lübeck, Germany. [3]Center of Mental Health, Hospital for Addiction and Addictive Behaviour, Klinikum Stuttgart, Germany. [4]Behavioral Law & Economics, Max Planck Institute for Research on Collective Goods, Bonn, Germany. [5]Institute for Cognition and Behavior, Vienna University of Economics and Business, Vienna, Austria. ✉ e-mail: soeren.krach@uni-luebeck.de

time spent on writing or evaluating funding proposals to carry out their actual research or education. Additionally, institutions allocate administrative resources to implement and manage the evaluation process. In some cases, the grant allocation process even depletes more resources from the system than they bring in, also known as the Szilard-point[12,13]. Peer-review-based funding schemes also exhibit biases against unconventional[2,14] or risky ideas[15] and systematically discriminate against people from historically marginalized groups[16,17]. Furthermore, with success rates at international funding agencies as low as 7%[18,19] and funding decisions depending on strokes of luck (e.g., drawing sympathetic reviewers, proposal aligning with current developments), traditional third-party funding schemes are increasingly resembling lotteries[1].

To address some of these issues, lotteries have been proposed as an alternative way of distributing funds[8,20–23]. While the random allocation of grants is seen as increasing the chances of marginalized scientists and unconventional approaches to receive funding, the scientific community has greeted this idea with skepticism, expressing fears of a threat to science due to a lack of quality control, as the gold standard of evaluation, peer review, is absent in a pure lottery[24,25]. Against the backdrop, a combination of grant lottery and peer review was advocated, i.e. lotteries were introduced as a tie-breaker at the end of the decision process in funding allocations (e.g., by the Swiss National Science Foundation, or the VolkswagenStiftung in Germany)[7,23,24,26–30] or to select applications after an initial screening (Explorer Grant from the Health Research Council of New Zealand)[31]. Under these systems, full applications are reviewed, and among the selected proposals, the final funding decision is determined by lot. However, while the tie-breaker lottery may eliminate certain biases in the last step of the decision process, two problems remain: First, the cumulative financial and temporal costs of the funding allocation process remain significant, driven by the many resource-intensive proposals for projects that ultimately will not be funded. Second, even if the lottery eliminates bias in the final decision-making step, it cannot prevent bias in the earlier stages of the procedure, including initial submissions. For instance, some applicants or groups of potential applicants may be more likely to submit proposals because they have more resources, better support structures, or more confidence than others[32,33].

The substantial individual and societal investments, combined with the egregious biases, have led to dissatisfaction with the current system[14]. Some have called for a radical change in the system, away from competitive funding distribution towards more base funding[34], while others advocate for stepwise improvements to current funding allocation processes[35]. In response, some third-party funding schemes have implemented a two-stage application procedure, initially reviewing only short proposals to reduce costs. The European Research Council (ERC) funding line requires an extended synopsis (5 pages) to be submitted alongside a full-length scientific proposal (14 pages). Only if the synopsis passes expert panel evaluation is the full proposal reviewed, leading to up to 65% of full proposals not being evaluated at all. Other funding lines, such as the VolkswagenStiftung, reduce costs not only for reviewers, but also for applicants: full-length proposals are invited only after a project is shortlisted based on evaluations of a short-form synopsis, with the full proposal requested in the second phase.

Recently, a more radical shift has been proposed[6,36], suggesting that lotteries prior to the peer review stage could be an effective strategy to address some of the remaining challenges in funding allocation. But would such a lottery-first approach improve the workload associated with proposal writing and evaluation, and reduce biases compared to other procedures? Empirical investigations of changes to funding schemes are largely lacking, difficult to implement[35], but crucial for generating evidence-based recommendations for policy changes in both private and federal institutions.

Here, we present data on an implementation of the lottery-first approach followed by peer review to allocate funding at a large German foundation (notably, the Danish Villum Foundation currently follows a very similar approach). We examine the changes in submissions and funded projects of female applicants after implementation, estimate the costs of the overall allocation process, and report on the attitudes and satisfaction of researchers and reviewers. Data were collected over 3 years as part of the Freiraum funding line at the German Foundation for Innovation in Higher Education (Stiftung Innovation in der Hochschullehre). The Freiraum call targets personnel from all disciplines within German higher education, including science management, and focuses on novel concepts for innovation in higher education. To reduce the direct and indirect costs of the funding model compared to relying on traditional peer review only, the initial funding round (2022) used a first-come, first-served procedure with peer review (see Methods for details). In the second and third funding rounds (2023 and 2024), a lottery-first approach was implemented, in which applicants first submitted a short expression of interest (1500 characters, outlining the project and giving personal details) and were then drawn in a lottery to submit a full-length proposal for subsequent peer review. During the most recent implementation in 2024 (overall budget €50 million for a maximum of 400,000 € per project over 24 months), we collected data from several waves of surveys with both applicants and reviewers (Fig. 1B, see https://osf.io/4ufrb/ for the original materials).

## Results

### Gender composition among research funding applicants and awardees

Prior evidence has demonstrated that traditional funding allocation is biased against certain groups, including women, non-native speakers, individuals from lower socio-economic backgrounds, early-career scientists, or people of color[37–41]. These disparities are evident not only in total funding volume—where men typically request and receive larger grants than women[42]—but in initial submission rates[43]. Specifically, the percentage of women submitting research grant proposals as principal investigators is lower than expected relative to their representation among eligible researchers[44]. How does a lottery-first approach[6] compare to a conventional approach in terms of female representation in research funding? To explore this question, we analyzed the gender composition of applicants and awardees in the three funding rounds of the Freiraum program. We compared the proportion of applicants identifying as female researchers in projects submitted and funded across the 2022, 2023, and 2024 funding rounds. Both the number of submissions from female scientists and the projects ultimately funded were consistently higher during the years the lottery system was employed. The proportion of female applicants increased from 40.8% in 2022 (first-come, first-served model) to 44.7% in 2023 ($\chi^2(1) = 3.92$, $p = 0.024$, one-tailed, OR = 1.19) and 45.1% in 2024 ($\chi^2(1) = 5.82$, $p = 0.008$, one-tailed, OR = 1.23, both lottery-first approach). These differences of 4.0 percentage points [90% CI: 0.6; 7.4] and 4.4 percentage points [90% CI: 1.0; 7.7] reflect relative increases in submissions by female applicants of 9.8% and 10.7%, respectively. This increase in initial submissions carried over to the final funding decisions. Women represented 38.0% of grant awardees in 2022, compared to 46.6% in 2023 ($\chi^2(1) = 2.81$, $p = 0.047$, one-tailed, OR = 1.43) and 47.4% in 2024 ($\chi^2(1) = 2.93$, $p = 0.043$, one-tailed, OR = 1.47). The respective differences of 8.5 percentage points [90% CI: 0.4; 16.7] and 9.3 percentage points [90% CI: 0.8; 17.8] correspond to relative increases of 22.4% and 24.5%. Overall, the proportion of female applicants and awardees was significantly greater under the lottery-first approach compared to the conventional first-come, first-served procedure.

### Costs of research funding

The financial and temporal costs associated with the conventional approach are substantial, largely driven by the preparation and peer

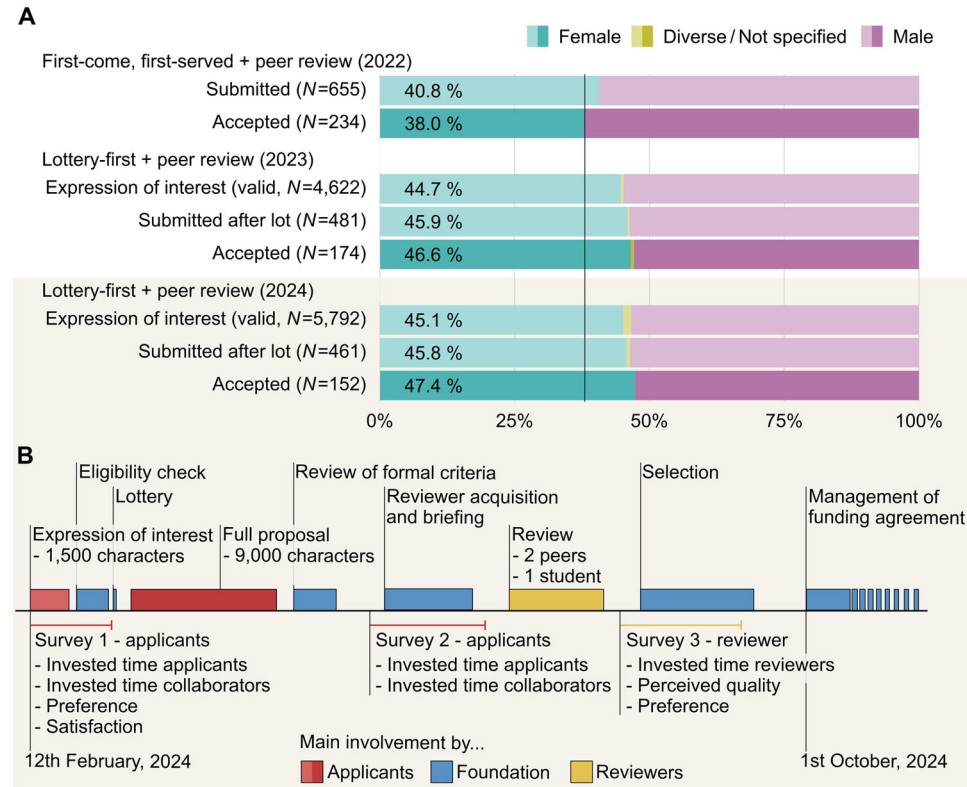

**Fig. 1 | Female representation across two different cost-efficient approaches to third-party funding and an overview of the surveys of this study during the lottery-first approach in 2024. A** Gender proportions at different stages for the Freiraum funding line over the past 3 years, since its establishment in 2022. This funding line was specifically designed to process applications in a cost-efficient manner. In 2022, funding allocation was based on a first-come, first-served procedure with subsequent peer review, where 40.8% of applicants were female, and 38% of the funded recipients were female. In 2023 and 2024, the allocation was conducted via a lottery-first approach with subsequent peer review. Under this approach, female researchers accounted for 45% of applicants and ~47% of funding recipients. See Methods for procedural details. **B** Timeline of the application and

evaluation process for the 2024 lottery-first approach. The process began with the submission of expressions of interest (maximum 1500 characters) on 12 February 2024, and concluded with notifications of funded proposals on 1 October 2024. A total of 6033 expressions of interest were submitted, of which 5792 (96.0%) met eligibility criteria. From these, 939 (15.6%) applicants participated in an initial survey. Following the lottery, 500 applicants were invited to submit full proposals (maximum 9000 characters), and 461 full proposals were ultimately received. After the full proposal submissions, 82 applicants participated in a second survey. Following peer review, 152 projects were selected for funding (success rate of 30.4% based on 500 winning lots). Among the 229 peer reviewers involved in the evaluation, 128 participated in a survey assessing the peer review process.

review of proposals that are ultimately not funded[8,45]. The lottery-first approach, by reducing the number of full proposals that need to be written and evaluated, offers significant potential for reducing costs and workload. Data from the Freiraum program demonstrate this potential. In 2024, a total of 6033 expressions of interest were submitted, and the target number of full applications was set at 500. Using self-reported estimates of working hours from applicants and reviewers (see Fig. 2A), the personnel hours allocated for managing the Freiraum funding line, and estimated hourly wages for the relevant stakeholder groups, we calculated the total personnel costs of the lottery-first approach. For 2024, these costs amounted to an average of €4.19 million across all simulation runs, representing ~8.4% of the total €50 million Freiraum funding volume for the year (Fig. 2C). In contrast, the estimated financial cost of a conventional funding approach instead of the lottery-first approach, where only a single stage of peer review is conducted for full proposals, was on average €13.13 million (based on self-reported confidence in submitting a full proposal, Fig. 2B).

Thus, the financial costs under the lottery-first approach are reduced by 68% compared to a conventional approach. Under the conventional approach, these costs would have consumed 26.3% of the total funding volume for this year. A breakdown of the cost estimates reveals that most working hours are dedicated to preparing full proposals (Fig. 2C), most of which are ultimately rejected

after review. The sunk costs for unfunded applications exceed those for funded applications by a factor of 26.7 in the conventional approach. In contrast, this ratio is lower in the lottery-first approach, at only 6.6 (Fig. 2D). Overall, the lottery-first approach is associated with substantially reduced estimated costs in developing and evaluating proposals, making the funding allocation process far more efficient.

## Satisfaction with the lottery-first approach
While the lottery-first approach is linked to increases in the proportion of female applicants and reductions in estimated costs, how do key stakeholders—applicants and reviewers—perceive it? Previous research on tie-breaker lotteries has shown that researchers favor a lottery step in later stages, as it encourages riskier proposals and reduces perceived bias[46]. Here, we provide evidence that a lottery as an initial step in funding allocation is also well-received by applicants. The number of expressions of interest submitted increased from the first implementation of the lottery in 2023 to the second in 2024, indicating no decline in engagement or shunning of the funding line after its introduction. Overall satisfaction with the funding line remained high (Fig. 3C). To further assess attitudes, we explicitly asked applicants and reviewers whether they preferred the lottery-first approach or a conventional funding approach (Fig. 3A). Applicants were also queried about their preferences regarding

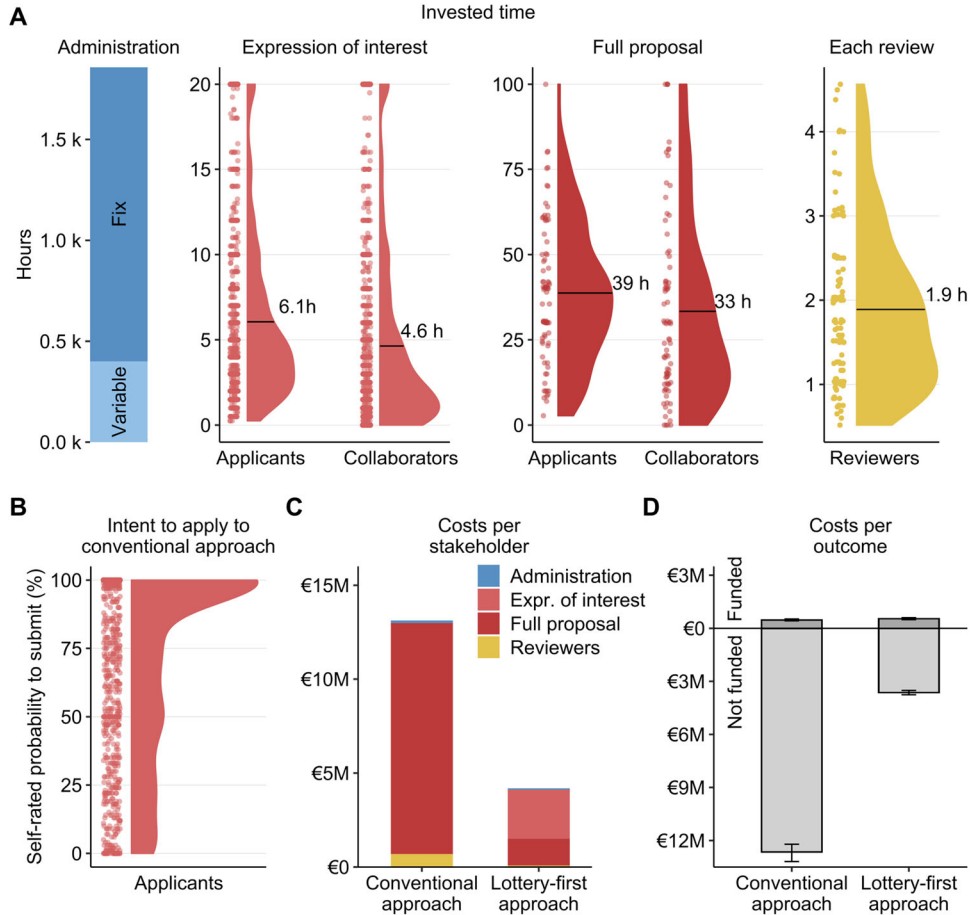

**Fig. 2 | Working hours and estimated costs for the lottery-first approach versus a conventional approach. A** Estimated working hours invested by various stakeholders during the application process for the Freiraum funding line, as implemented by the Stiftung Innovation in der Hochschullehre in 2024. Administrative working hours are divided into fixed costs for managing processes and variable costs that scale with the number of submissions at each stage. Applicants' self-reported working hours are derived from survey responses. For the initial expression of interest, applicants estimated an average preparation time of 6.1 hours (h) (±4.8 hours, standard deviation), with collaborators contributing an additional 4.6 hours (±5.3 hours). For the full proposal stage, applicants reported an average preparation time of 39 hours (±20 hours), while collaborators invested 33 hours

(±27 hours). Reviewers reported an average of 1.89 hours (±0.97 hours) to evaluate a single full proposal. **B** Applicants' confidence in continuing their projects as full proposals without an initial lottery, as assessed in the first survey. Confidence levels were measured on a scale from "definitely yes" (100%) to "definitely no" (0%). **C** Estimated personnel costs for a conventional approach, where full applications are submitted directly to peer review, compared to the lottery-first approach. Costs are calculated based on the working hours invested and are split by stakeholder groups for $n = 1000$ simulation runs. Expr. of interest: Expression of interest. **D** Total estimated resource investments for both funded and unfunded applications for $n = 1000$ simulation runs (data are presented as mean values, error bars indicate minima and maxima).

workload, perceived chances of success, fairness, and the expected quality of funded applications. Among applicants, 49.6% ($n = 384$) expressed a preference for the lottery-first approach, 2.6% ($n = 20$) had no preference, and 47.8% preferred the conventional approach. Applicants particularly favored the lottery-first approach regarding reduced workload (Fig. 3B). However, the conventional approach was preferred for perceived chances of success, fairness, and the expected quality of funded applications. Reviewers exhibited a more divided perspective, showing strong preferences for either the lottery-first ($n = 49$, 52.1%) or conventional approaches ($n = 45$, 47.9%). Importantly, the assessed quality of the proposals was in line with the reviewers' expectations, with no indication of unusually low-quality submissions on average (Fig. 3D). Concerns that the quality of proposals submitted to peer review might suffer from the implementation of a lottery-first procedure were therefore mitigated. In summary, the perceptions of applicants and reviewers regarding the lottery-first approach were broadly comparable to those of a conventional funding allocation process, suggesting it is a viable alternative for funding distribution.

## Discussion

Current third-party funding allocation models for research and higher education face criticism for their high costs and biases against certain groups (e.g., gender bias)[35,47]. Using data from three years of funding allocations and complementary surveys, we provide empirical evidence that a lottery-first approach is associated with an increased female representation both at the submission stage and among funded projects, lower estimated costs, and is appreciated by a substantial proportion of applicants and reviewers.

The lottery-first approach has the potential to enhance funding opportunities for structurally disadvantaged groups. Dedicating significant working hours to grant applications can be a substantial risk for researchers and higher education employees. This is particularly pronounced for women, who often face disadvantages due to precarious employment conditions (e.g., lack of long-term contracts, limited access to mentors, inadequate resources, and constrained flexibility)[43,48]. Career disruptions caused by life events such as childcare or illness[49], also disproportionately affect women and other structurally disadvantaged and historically marginalized groups[3,4,39].

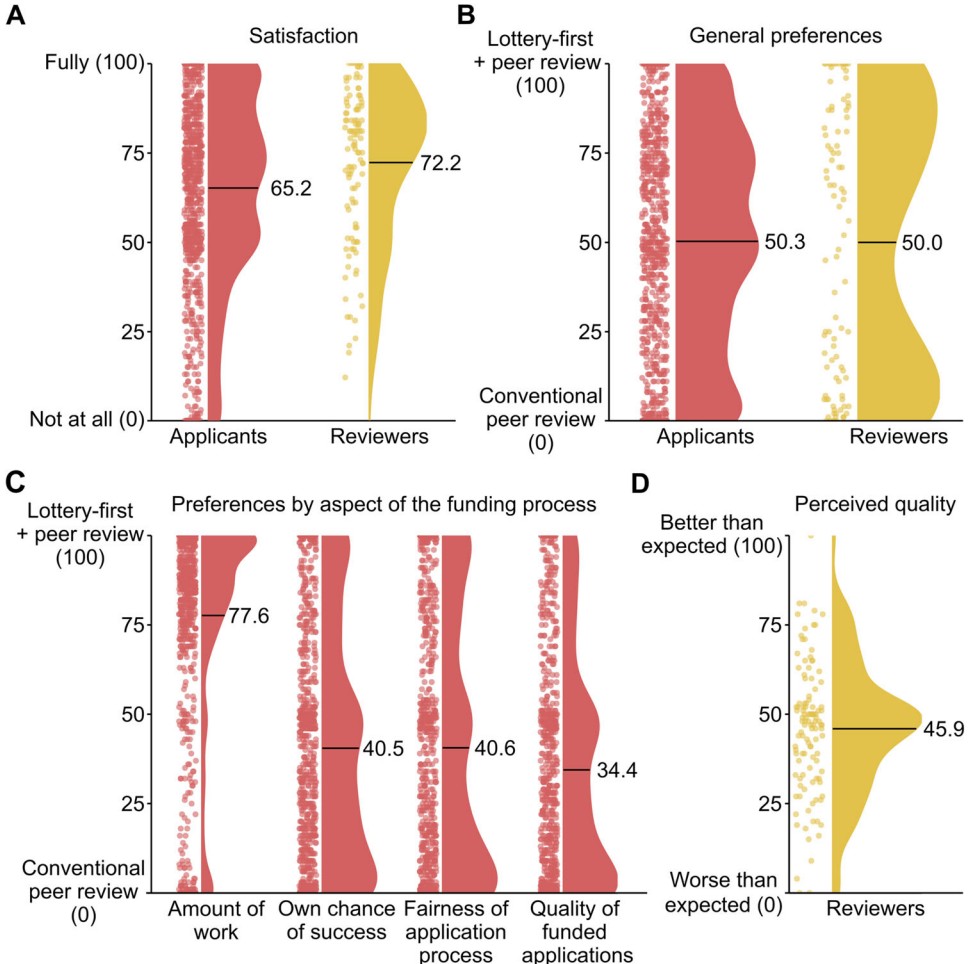

**Fig. 3 | Preferences and satisfaction with the lottery-first approach. A** Overall satisfaction with the Freiraum funding line in 2024. Satisfaction was high among applicants after submitting their expression of interest (mean = 65.2, SD = 26.6) and among reviewers after completing their evaluations (mean = 72.2, SD = 21.59; see Supplementary Table 4). **B** General preferences for a lottery-first approach compared to a conventional approach under similar acceptance rates. Applicants' preferences were mixed, with 49.6% favoring the lottery-first approach, 2.6% reporting no preference, and 47.8% preferring the conventional approach (mean = 50.3, SD = 31.2). Reviewers' preferences were similarly divided, with 52.1% preferring the lottery-first approach and 47.9% favoring the conventional approach (mean = 50.0, SD = 37.8). **C** Applicant preferences regarding different facets of the

funding process. Applicants strongly preferred the lottery-first approach for reducing the amount of work involved (mean = 77.6, SD = 29.6; 83.7%, *n* = 691 in favor). However, they expressed a preference for the conventional approach in terms of perceived personal chances of success (mean = 40.5, SD = 33.2; 33.4%, *n* = 266 in favor of the lottery-first approach), fairness of the application process (mean = 40.6, SD = 36.0; 35.1%, *n* = 278), and the quality of funded proposals (mean = 34.4, SD = 31.1; 24.5%, *n* = 183). **D** Reviewers' perceptions of the quality of submitted proposals. Reviewers reported that the quality of applications was, on average, in line with their expectations (mean = 45.9, SD = 18.1, measured on a scale from 0: "worse than expected" to 100: "better than expected").

The time spent preparing grant applications diverts effort from other critical responsibilities, including research, teaching, and supervision, with no guarantee of success. Introducing short expressions of interest as initial submissions in a lottery-first approach can lower entry barriers during the application process. Once selected through the lottery for submission of a full proposal, applicants benefit from a higher success rate compared to conventional funding allocation processes (approximately 30% in Freiraum). The Freiraum data demonstrate that the proportion of submissions and funded projects led by female applicants was consistently higher under the lottery-first approach, suggesting that the choice of funding allocation method may influence who applies and who does not.

The Freiraum data further demonstrate that the estimated costs associated with preparing, evaluating, and administering funding applications are substantially reduced under a lottery-first approach. The cost-saving potential of this approach is particularly notable because its expenses are minimally influenced by the number of applicants, scaling instead primarily with the number of lottery tickets.

This makes costs more predictable from the outset, simplifying budget planning for funders, especially for smaller organizations or new funding lines where the number of applicants is difficult to anticipate.

Simultaneously, growing concerns question whether the benefits of conventional funding allocation procedures justify their high costs. On one hand, reviewers' ex-ante assessments are often less reliable than desired[50,51], and biases persist in the evaluation process regardless of proposal quality[52]. On the other hand, as shown here, the overall costs of preparing and reviewing applications in a conventional system are substantial and may even exceed the total funding budget to be allocated[13]. Much of this expenditure is wasted on preparing and reviewing proposals that are ultimately rejected.

Although peer review remains the gold standard for evaluating proposals, streamlining funding allocation processes with additional mechanisms like lottery-first approaches could unlock resources for the academic system in several ways. First, the time saved could be reallocated to conducting more, and potentially higher-quality, research and teaching. Second, fewer applications requiring review

due to an initial lottery could allow for improvements in peer review processes compared to the traditional approach. Resources could be redirected toward reviewer training[53], implementing debiasing interventions[54,55] and better familiarizing reviewers with unconventional proposals. Additionally, more reviewers could be assigned to each application, increasing the reliability of evaluations and better reconciling divergent opinions, particularly for innovative or groundbreaking ideas[56]. Since an acceptable reliability requires many more reviewers than is standard practice, with some estimates going as high as ten[57] or twelve[58], having fewer applications to review could enable a much higher-quality selection.

The lottery-first approach offers a viable alternative to the increasingly common practice of limiting the number of applications permitted from a single institution to reduce the workload of funding agencies. Such a limited submission approach can introduce unintended and highly idiosyncratic biases during the institutional preselection process, potentially affecting which projects are advanced for submission[59]. Furthermore, while this strategy reduces administrative costs for funding agencies, these costs are often shifted to the academic institutions tasked with preselecting applications. As a result, limiting submissions at the institutional level not only risks adding bias to the selection process but also proves to be less effective in reducing overall costs compared to the lottery-first approach.

The Freiraum data indicate that the lottery-first approach was more popular among applicants and reviewers than anticipated. This suggests that a significant number of applicants may be dissatisfied with the current funding system and that there is support within the academic community for changes to third-party funding models. Applicants clearly weigh the advantages and disadvantages of different funding allocation models, with preferences for the lottery-first approach driven primarily by its reduced workload. This finding implies that conventional approaches offer insufficient returns on the investments of time and effort they require. Notably, this preference for reduced workload outweighs applicants' perceptions of lower chances of success, quality, or fairness associated with the lottery-first approach, resulting in a preference distribution in which neither the lottery-first approach nor the traditional peer-review-only process is clearly preferred.

One argument that is often brought forward against the lottery-first approach is a presumed lower quality of funded proposals. There are several reasons against this assumption: First, reviewers' quality assessments indicated that proposals submitted under the lottery-first approach were generally as strong as expected, allaying concerns that this method might significantly compromise the quality of proposals or subsequent research. Second, it is true that in a lottery-first approach, the quality of an application does not affect its probability of advancing. However, the extent to which this leads to a reduction in the quality of funded applications compared to the traditional approach depends on the accuracy of the peer review in both approaches, since traditional approaches are also not error-free. Given biases and unsystematic errors in peer review, the often very short initial applications in traditional approaches, and higher entry biases[51,60,61], it is not certain but rather an empirical question whether the lottery-first approach, on average, favors lower-quality proposals than a non-ideal traditional one. Probabilities of advancing are very easy to calculate in lottery approaches, but very difficult to estimate for peer review. For this reason, in a previous publication, we provided an interactive tool to compare different approaches in terms of quality, cost and fairness[6,62] and called for more empirical research on grant quality. Overall, each funding model has its advantages and disadvantages, and opinions on the best model vary widely among researchers. These might also depend on the amount of funding and the goal of the funding program. In the present case, we analyzed a funding program for advances in higher education, where acceptance rates of a lottery-first approach might differ compared to a funding program for basic research. In general, diversifying the funding landscape and exploring innovative options can provide a productive path forward for ensuring unbiased and efficient resource allocation, something that is also desired by the applicants[24]. Achieving this requires further empirical investigations to generate robust, evidence-based recommendations for practical implementation in funding programs, especially for research grants. Such evidence should encompass long-term, high-quality evaluations of various types of projects funded under different allocation mechanisms, as well as experimental research to inform evidence-based reforms and policies that address trade-offs between cost-effectiveness, bias, and quality[46].

Notably, these results come with some limitations. Importantly, the case of the evaluated funding program specifically targets innovative teaching approaches, rather than research projects. This impacts the generalizability of the provided cost estimates and the preferences for the lottery-first approach. As teaching grants are rare, applicants have fewer options when it comes to applying for funding. Thus, the conventional approach could be especially costly in the context of teaching, since a higher proportion of potential applicants would need to apply, creating high demand for reviewers and low success rates for applicants. This highlights that the actual cost-saving potential of a lottery-first approach depends on the specific scenario of the funding program, including funding demand and the characteristics of applications, such as the number of hours invested in submitting them. Along these lines, it is also reasonable to assume that the preference for the lottery-first approach is proportional to the number of work hours invested in a potentially rejected application, as well as to the applicant's demand for fairness, in the sense of selecting the best submissions rather than one of the best from the applicant pool in the funding program. This may differ for research projects. Relatedly, representation could only be estimated regarding gender[63], and also only compared to the approach previously used by the foundation. Either might not be representative of different types of representation and funding lines. However, it is safe to assume that a validly conducted lottery is bias-free, and thus any biased selection process replacing part of its process with a lottery will have less bias. Thus, the question is less about quantifying bias reduction than whether the advantages and disadvantages of a specific selection process in combination justify its use. One important aspect of this is the quality of proposed projects. Here, we were not allowed to assess the full text of the proposals, and we were thus not able to estimate their quality. Since proposal quality is potentially still different from the importance and quality of research projects, future research should therefore compare different funding scenarios from submitted proposals to fully implemented projects.

The observational data presented here suggest that lottery-first approaches hold promise as an innovative funding mechanism. This approach is associated with increased female representation and reduced estimated costs, while remaining acceptable to both applicants and reviewers. It also offers more controllability for funders since they can determine the number of reviews in advance by deciding on the number of winning tickets. By rethinking how funding is distributed—such as by incorporating lottery-first mechanisms—funders have an opportunity to design research systems that generate robust knowledge, foster innovation, address pressing global challenges, and maximize the collective benefits of high-quality scientific inquiry.

## Methods

All analyses and data visualizations (except Fig. 1B) were performed using R[64] with packages from the tidyverse[65] and patchwork[66]. For a detailed list of all packages, including version numbers please see the accompanied material on the OSF.

### Background on the funding line

The Foundation for Innovation in Higher Education (Stiftung Innovation in der Hochschullehre; StIL) administers different funding

programs on behalf of the German Federal Ministry of Education and Research, with an annual budget of €150 million. The funding lines are interdisciplinary in nature and are designed to enhance teaching in German higher education. The funding line Freiraum ('Open space'), described here, aims to provide successful applicants with the resources to develop and evaluate their ideas for enhancing teaching practices without any specific thematic or subject focus. Projects are selected based on their potential to innovate teaching in higher education in Germany (see Freiraum–Stiftung Innovation in der Hochschullehre). In Germany, >280,000 people from all higher education institutions were eligible to apply for this funding line, including staff working in research and teaching (professors, postdocs, and doctoral students) and academic management staff. Overall, the funding line stands out in the German context due to its broad eligibility criteria and large funding volume dedicated to higher education. Comparable funding programs are available from the StIL itself (€330 million dedicated to digitalization in higher education), while other funding for innovation in higher education is usually allocated via smaller-scale funding lines.

From the beginning, procedures for submitting to the funding line were designed to save costs in the allocation process. Two different allocation mechanisms were used: First, when the funding line was established in 2022, it followed a first-come, first-served procedure for a total of 600 proposals (including, among others, a description of the problem or state of the art, an outline of the project plan including objectives and work plan, description of the project's contribution to developing the state of the art, and financial details). Since proposals could involve collaborations between multiple principal investigators (PIs) from different institutions, in the end, proposals of 655 individual applicants were submitted before the submission portal was closed. Second, in 2023 and 2024, access to the funding line was changed to a lottery-first approach. Applicants first submitted a short expression of interest (1500 characters) in which they outlined the scope of their project and provided personal details, including name, gender, academic titles, e-mail address and phone number for correspondence, institution, personal role at the institution, and subject area ($N = 4918$ in 2023 and $N = 6033$ in 2024). The expressions of interest were checked by the StIL for formal eligibility criteria: Specifically, identical or very similar expressions of interest from the same work group were excluded from the lottery. After checking these criteria, valid proposals of 4622 applicants in 2023 and 5792 in 2024 were subjected to a lottery (lottery-first approach) to select 500 proposals. This resulted in 534 individual applicants in 2023 and 500 in 2024, who were invited to submit a full proposal (9000 characters) for review. In 2024, each PI was required to submit a proposal individually, even in cases where collaborations between institutions were planned, so that the number of selected proposals matched the number of applicants (this was not the case in 2022 and 2023). Applicants had seven weeks to submit their full proposal (481 applicants in 2023 and 461 in 2024 submitted proposals). In all three application periods, both expressions of interest (only in the years 2023 and 2024) and the full proposals were first formally evaluated (e.g., formal checks, correspondence between expressions of interest and full proposals where applicable) and then subjected to peer review. Each application was reviewed and scored by two peers and one student in 2023 and 2024, and one peer and one student in 2022. The final selection of funding applications was made by a selection committee managed by the StIL, based on the peer-review evaluation. Overall, 234 applicants were funded in 2022, 174 in 2023, and 152 in 2024.

**Gender proportions in funding periods.** We compared the proportion of female applicants across the different funding periods and stages. For this purpose, we received anonymized data from StIL on the gender of applicants in the 2022, 2023, and 2024 funding periods. For every dataset, the applicants' gender was coded based on the available data from the formal assessment during the application

process. Each applicant's gender was classified based on the German salutation that applicants had selected in the respective forms of the submission process: 'Herr' ['Mr'], 'Frau' ['Ms'], and 'keine' ['not specified']. In case no salutation was specified, gender was classified based on gender self-assessments within the application forms in case they were available (i.e., 'weiblich' ['female'], 'männlich' ['male'], 'divers' ['diverse'], 'keine Angabe' ['not specified'], '---'). Based on the available data, 'Ms' or 'female' was classified as 'female', 'Mr' or 'male' was classified as 'male', and all other categories were classified as 'diverse/not specified'. This classification was reliable, as there were no conflicting gender classifications based on the different self-report sources (salutation or 'gender'). While we acknowledge that the chosen approach to defining gender may be overly simplistic, the present analyses are necessarily limited to the available historical data. Thus, we also restricted our statistical tests to the comparison between male and female applicants and ran one-tailed proportion tests, comparing the initial application phase and the funded application within each year the lottery was implemented against the year without a lottery (2022). We decided to use one-tailed tests as we did not assume that female applications and awardees would be reduced under the lottery approach compared to the first-come, first-served model.

**Evaluation of the 2024 funding period.** In addition to the data we received from StIL, we conducted an independent evaluation through several surveys during the application process in 2024 to determine the investment in personnel, the attitudes of the participants and the reviewers, and the quality of the applications. Participation in the surveys was voluntary. The online surveys were generated and made available using SoSci Survey[67]. The survey data were only accessible to the authors, not to the StIL. Invitations to the surveys were sent via e-mail by the StIL to all eligible applicants after the submission of (i) the expression of interest, (ii) the full proposal, and to the reviewers after the collection of reviews (Fig. 1B; two additional invitations were sent after the draw and after the final recommendation). Each survey was similarly structured, with applicants receiving instructions and providing informed consent at the beginning. Participation was voluntary, not linked to participants' proposals or reviews, and participants received no compensation. Sample sizes resulted from the convenience sample available. No statistical method was used to predetermine sample size. All surveys were conducted under block ethics approval to the MPI Decision Lab (RR) granted by the Ethics Council of the Max Planck Society to perform experimental research with human participants following standard procedures, including anonymized data collection and procedures without deception. Under this block approval, survey data for the analysis were obtained. StIL data (i.e., information on the number of submitted or funded proposals by gender) was collected under the data protection agreement of the StIL submission portal and was provided at the discretion of the foundation and in accordance with the data protection agreement applicants agreed to when registering on the StIL platform. The first survey (survey period from 12 February to 6 March 2024) targeted all applicants who had submitted an expression of interest. Altogether, $N = 939$ applicants provided informed consent and started the survey (response rate of 15.6%). For the data analyses, $N = 45$ invalid cases were removed (see handling survey data), leading to a sample size of $N = 894$ valid cases. Sample characteristics are provided in the Supplementary Table 1. The survey assessed attitudes toward funding approaches, personnel investments, subjective confidence in submitting the application as a full proposal for review without a lottery, and overall satisfaction with the funding program. Overall, although the survey response rates were rather low, they can be considered typical for current surveys of busy academics. Notably, there could be a distortion within the responders in the way that only highly motivated applicants participated in the survey and therefore may skew the results. This could, for instance, partly explain the bimodal distribution of reviewer preferences.

The attitudinal data we present here relate to preference for one of two approaches to funding allocation: the lottery-first approach and the conventional approach to submit a full proposal to peer review directly. The two funding scenarios were described as follows: "In the lottery-first approach, applicants first submit an expression of interest (1500 characters). Second, ~10% will be selected by lot. Third, those selected are invited to submit a full application (9000 characters). Fourth, the full proposal is reviewed by peers, with the rate of funded projects estimated at ~40% after the review. In the conventional approach, applicants first submit a full proposal (9000 characters). The applications are then reviewed by peers, with the rate of funded projects estimated at ~4% after the review." Both approaches thus result in a similar number of winners.

Preferences for one of the funding lines were expressed on a horizontal visual analog scale, with the lottery-first approach on one end and the conventional approach on the other end of the bipolar scale (for the purpose of the analyses, full preference for the lottery-first approach was coded as '100' and full preference for the conventional approach as '0'). Applicants could also decide not to provide an answer, which was the default option. In addition to the general preference for one of the funding scenarios, various other aspects that could influence the general preference were also assessed on the same scale. Participants indicated preferences about their own investments, their own chances of success, fairness, and the quality of the funded applications.

Satisfaction with the funding line was assessed by the question 'How satisfied are you with the Freiraum funding line?', where applicants provided responses on a visual analog scale ranging from 'not at all satisfied' to 'very satisfied' (for the purpose of the analyses, 'very satisfied' was coded as '100' and 'not at all satisfied' as '0'). The data on investments in personnel reflect the invested work hours for submitting the expression of interest by (1) the applicant and (2) the collaborators involved (co-applicants or staff not explicitly mentioned in the proposal). Applicants provided estimates on a visual analog scale ranging from '0 hours' to '20 hours or more', with stepwise increments of 15 minutes. They were asked to estimate roughly how much time they had spent, including on conceptualization, discussions, and writing the 1500-character expression of interest, and how much time the collaborators had invested in total. Applicants' confidence in submitting a full proposal without a prior lottery draw was assessed as follows: "Would you have submitted a full application (9000 characters) directly without first submitting an expression of interest?" Answers were provided on a visual analog scale ranging from 'definitely not' (0) to 'definitely yes' (100).

The second survey (survey period from 23 May to 30 June 2024) targeted all applicants who had submitted a full application. A total of $N = 82$ gave their consent and answered at least one of the questions asked (response rate of 17.8 %). For the data analyses, one invalid case was removed (see handling survey data), leading to a sample size of $N = 81$ valid cases. Sample characteristics are provided in the Supplementary Table 2. The question format on the hours spent working on the full proposal was like in the first survey. Only the scale anchors for the self-invested working hours were adjusted from '0 hours' to '100 hours or more' to meet the increased length requirements in this phase.

The third survey (survey period from 24 July to 31 August 2024) targeted all 229 reviewers who completed their reviews. Altogether, $N = 129$ responses were collected (response rate of 56.3%). For the data analyses, one invalid case was removed (see handling survey data), leading to a sample size of $N = 128$ valid cases. Supplementary Table 3 provides detailed information on the status groups of the reviewers. This survey assessed the staff hours invested in reviewing the individual full proposals, the attitude towards the funding line, and the quality of the applications. To estimate the number of hours spent on the review, reviewers were asked to estimate, on a visual analog scale, how much time they had spent on average on a review, ranging from '0 minutes' to '300 minutes or more' with 1-minute steps. The quality of the grants reviewed was assessed using the question 'The quality of the

applications meets my expectations.', which was evaluated using a visual analog scale with the anchors 'worse than expected' (coded as '0') to 'better than expected' (coded as '100'). Satisfaction with the funding line and the general preference for a lottery-first approach or a conventional peer review were assessed with questions identical to those asked of applicants in the first survey.

**Cost estimations.** Personnel costs of a lottery-first approach compared to a conventional peer review approach were simulated 1000 times based on estimates of the working hours of the staff at the StIL and the survey data. Detailed estimates of the StIL for the foundation's working hours are provided in the Supporting Data of Fig. 2, which breaks down the fixed costs and workload depending on the number of submissions at the various stages of the submission process. The calculation of the potential costs of a conventional peer-review approach was conducted as follows: First, it was assumed that all initial 6033 Freiraum applicants are potential applicants in a conventional peer review approach, but that not all would apply due to the higher initial resource investment. Thus, whether each of the 6033 applicants would have submitted a full proposal was determined by drawing a respondent's certainty to submit a full proposal as rated on the visual analog scale in survey 1 (0–100%) as input for the probability in a Bernoulli distribution. Second, if this draw resulted in a decision to submit a full proposal, the applicant's working hours and the collaborators' corresponding working hours were drawn from the respective responses in survey 2. If this draw resulted in a decision not to submit, no working hours were counted for this applicant. Third, the working hours for reviewing the full proposal were determined by two draws from the distribution of the invested work hours of peers in survey 3, excluding student reviewers since they received fixed compensation (see below). Fourth, the sum of the working hours was multiplied by €40, the gross costs per working hour for staff employed at universities in Germany. According to official figures from the German government (Arbeits- und Lohnnebenkosten−Statistisches Bundesamt), this reflects a rather conservative estimate for personnel costs. Fifth, a compensation for the student reviewers was added: They received €15 from the StIL for each review. Finally, fixed and variable personnel costs at the StIL were added for general management, handling the submissions, and selection and management of the funded applications.

To estimate the costs of the lottery-first approach, we ran equivalent simulations. Here, all 6033 applicants submitted an expression of interest, and we estimated the working hours for preparing the expression of interest based on a random draw of the respective responses in survey 1, as depicted in Fig. 2A. Here, we analyzed responses from participants who provided complete data on both applicants' and collaborators' working hours, excluding those with missing values in either variable. Additionally, if participants indicated that they had no collaborators but did not respond to the question regarding the time collaborators spent, this variable was set to zero (six responses). For the next step, one draw was made from the distribution of working hours for the full proposals for each of the targeted 500 submitted applications and two draws were made from the distribution of working hours of the reviewers. The overall sum of working hours was multiplied by €40, and the compensation for the student reviewers of €15 was added for each submitted full proposal. Finally, the fixed and scalable costs at the StIL for the handling of the process, including the general management, handling of the expression of interest, full proposals, selection and management of funded applications, were added. In both scenarios, the lottery-first approach as well as the conventional approach, 150 proposals were eventually funded for each of the 1000 runs and their costs were compared to the costs of the rejected proposals. Note that because cost estimates are based on self-reported working hours, they may differ from the true value due to systematic reporting biases. In addition, we have not modeled potential sources of variation between participants, such as whether they could reuse

materials from an earlier unsuccessful proposal, their available time budgets, or their career status.

**Handling survey data.** It was necessary for participants to run Java-Script in their web browser to ensure the correct display of the visual analog scales. This information was conveyed in the instructions provided to the participants, and a technical query was used to ascertain whether the instructions were adhered to. Due to this query, the data sets of seven applicants in the first survey and one participant in the reviewer survey could not be included in the analyses. In addition, applicants with excessively fast relative response times were excluded from the analyses, as these reflect invalid response behavior, in accordance with the SoSci survey recommendations[68]. This led to the exclusion of 38 applicants in the first survey and one applicant in the second survey. For each question, the sample size is given for all valid responses, with missing responses excluded.

### Reporting summary
Further information on research design is available in the Nature Portfolio Reporting Summary linked to this article.

## Data availability
The raw data from the Stiftung Innovation in der Hochschullehre, as well as the survey data, are protected and are not available due to data privacy laws. All processed and anonymized data are available at https://osf.io/4ufrb/ (https://doi.org/10.17605/OSF.IO/4UFRB). Source data are provided with this paper.

## Code availability
The R code to reproduce the analyses and figures is provided (https://doi.org/10.17605/OSF.IO/4UFRB).

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

## Acknowledgements

We would like to thank the Stiftung Innovation in der Hochschullehre for the cooperation, especially Marieke Rother, Simone Gobien, Evelyn Korn, Leon Bleser, Merle Siefert, Felix Schimmelpfennig, and Cornelia Raue. We would also like to thank Thomas Martinetz and Daniel Rapoport for their critical input on an earlier version of the manuscript. Finally, we would also like to thank the reviewers for their very helpful reviews.

## Author contributions

S.K., F.L., and L.R. established the collaboration with the Stiftung Innovation in der Hochschullehre. F.L., S.K., F.M.P., L.R., and R.R. developed the surveys. F.L. and L.R. programmed the surveys. F.L., L.R., and F.M.P. analyzed the data. F.L. and F.M.P. created the figures and tables. All authors drafted the first version of the manuscript, developed the figures, and approved all changes in the final manuscript. Apart from the first author, F.L., the other authors are listed in alphabetical order as follows: S.K., F.M.P., L.R. and R.R.

## Funding

## Competing interests

The authors declare no competing interests.
