## [Transparent Peer Review file · Nature Communications]

Lottery before peer review reduces gender bias and economic cost in research funding allocation

Corresponding Author: Professor Sören Krach

Version 0:

Reviewer comments:

Reviewer #1

(Remarks to the Author)

This is a genuinely exciting paper that examines a big shake-up to research funding. I was aware of the same idea being currently be applied in Denmark (<https://villumfonden.dk/en/group/grantsubarea/villum-experiment>) so it was a pleasant surprise to see results earlier than I expected. I congratulate the funders for trying this innovation, as this is the only way to improve the system.

The results were very interesting with enormous reductions in costs, which is a potentially huge benefit as many funding systems have become over-competitive and inefficient.

The study design was straightforward and appropriate.

A result that I think needs more discussion is the enormous increase in applications after the policy change. I think this should be mentioned in the abstract, as total application numbers are a key concern for funding staff. The numbers went from 655 submitted to 4,622 and then 5,792, which is an enormous increase. I appreciate that the lack of difference in quality scores is comforting, but there are other potential problems.

One potential issue is “stalking horse” applications, where several applications are submitted to the lottery in the hope of getting at least one ticket. For example, a lab-head may tell all their postdocs to submit vaguely worded applications, then they take-over any winning ticket with their own idea. There are potential issues here with bullying and misrepresentation. I note that 19 and 39 proposals of the randomly selected 500 did not submit a proposal (line 477) and perhaps these were spare slots that were no longer needed when a lab won 2 or more tickets?

Thinking more positively, the large increase in applications could good news as more ideas are being gathered from the community. This may increase the diversity in ideas, which was a key point made in Shahar Avin’s PhD thesis (reference number 11 on page 14).

Minor comments

- “exclusion” is too strong a word (in my opinion) when talking about the bias against minorities. There has certainly been discrimination, but exclusion implies that there were no winners from these groups.
- The sentence starting in line 178 came too early, as the difference in costs was only made clear in the next paragraph.
- 39 hours (on average) seems relatively low for a full application and based on the list of what applicants had to provide (lines 462 to 464). I note that the question was worded in terms of hours (line 553) and the results may have been different if the scale was in days.
- The almost 50:50 split in preference is fascinating (line 229).
- Line 300, when the application costs exceed the available budget, this is known as the Szilard point, see https://en.wikipedia.org/wiki/Szilard_point
- Lines 522, 564, 571, the survey response percentages can be rounded and do not need 2 decimal places.
- The low survey response rates should likely be mentioned in the limitations as well as the likely direction of any bias. The response rates are typical for current surveys of busy academics.
- I would add the average grant size per successful application. This will be one of the first questions for any funder looking to implement a similar scheme.

(Remarks to the Author)

The article deals with an important and much sought-after topic. Within the community of research funders and science policy makers, there is a great deal of interest in simplified grant distribution, including the early use of lotteries. For this reason, the results of the lottery-first approach presented here are likely to meet with great interest.

Against this background, it seems to me particularly important that the article in its current form is rewritten to include further studies on lottery procedures in research funding. I think this is extremely important because the results presented need to be seen and discussed in the context of other findings. The article is written in a style typical of the journal *Nature*, but the results presented are not original to the same extent. This does not detract from the importance of the results presented, but does call for a stronger contextualization of the findings.

I will list a few points and recommend that they be taken into account in the revision:

1) In some cases, the authors cite literature but do not fully consider its findings. For example, they cite Liu, M. et al. (2020) in the journal *Research Integrity and Peer Review* as a source for the Health Research Council (HRC). However, Liu and colleagues do not just describe the use of a lottery in the HRC funding process. They report on how applicants view the use of their lottery and what they accept and prefer. This finding strengthens the case for a particular type of lottery. The current paper is therefore not the first to investigate the acceptance of lotteries by researchers. The results should therefore be seen in the context of other findings. In particular, the authors should consider two surveys of the German scientific community that report scepticism and substantial rejection of random grant allocation. For example, the DZHW survey of scientists in 2019 shows that 69 percent of participants reject a random distribution of funding if the proposals are formally correct (Ambrasat, J. and Heger, C. 2020 in the report *DZHW Barometer für die Wissenschaft*). This finding is replicated in another survey of scientists in 2021, which shows that 66 percent reject a purely random distribution of grant proposals (Philipps, A. 2022 in the journal *Science and Public Policy*). On the other hand, respondents to the Liu et al. and Philipps surveys support procedures that combine lotteries and peer review. This is also consistent with the observation by Philipps (2021 in the journal *Research Evaluation*) that German scientists who deliberate about the random distribution of grant proposals emphasise the importance of scientific institutions such as peer review. Peer review may be problematic in some respects, but it is considered the gold standard in science for checking the originality and feasibility of proposed research. In my view, this seems relevant to understanding the responses to the lottery-first approach in the survey reported here. The lottery-first approach presented is a combination of lottery and peer review (see line 530-537), and my reading is that this aspect is important to the participants. They are open to the lottery, but at the same time a large proportion of them is still in favour of conventional peer review and they may support the lottery-first approach only because it includes a peer review process in the second round.

2) In addition, the authors cite Gross and Bergstrom (2019) in the journal *PLoS Biology*, but do not consider similar research. There are other studies that show how efficient a lottery approach can be (including acknowledging that random distribution is less costly, increases diversity, and increases opportunities for minorities). I provide this list of research because the results of the present study support what has been shown previously (especially regarding less work and cost):

Avin, S. 2015 in *Recent Developments in the Philosophy of Science*.

Avin, S. 2018 in *RT. A Journal on Research Policy and Evaluation*.

Bieri, M., Roser, K., Heyard, R., et al. 2021 in *BMJ Open*.

Höylä, T., Bartneck, C., and Tiihonen, T. 2016 in *Scientometrics*.

3) In light of other research on lotteries and research funding, the authors should also reflect more on the "Freiraum" funding line. This initiative is aimed at scientists and science managers. It is not about basic research but about innovative concepts for higher education, i.e. it is not about potentially risky research. Rather, the concepts proposed are potential solutions based on proven knowledge. This may have an impact on who submits to "Freiraum". The focus on one particular funding initiative also limits the results. This limitation should be stated in the paper.

4) The paper also gives no information about the amount of money an approved proposal might receive. I would argue that the acceptance of a random grant allocation depends on the size of the grant amount.

5) Lines 326-327 state: "Notably, this preference for reduced workload outweighs applicants' perceptions of lower chances of success, quality, or fairness associated with the lottery-first approach." What is meant by "outweighs"? Reduced applicant workload is a positive of lotteries, but it is not comparable to chances of success, quality, or fairness. It is clear that a lottery reduces the workload (see for example Shaw, J. 2022 in the journal *Research Evaluation*), but it is less clear that it addresses other shortcomings of peer review. However, scientists seem to be concerned about the quality of proposals when proposals are selected randomly, and established scientists in particular are sceptical because a lottery reduces their chances of success compared to a conventional peer review process (see Philipps 2022). How do the results presented agree or disagree with previous results?

5) I was confused reading that "fewer applications requiring review could allow for improvements in peer review processes." (line 304) Of course, reviewing 481 full proposals in 2023 is less than the 655 in 2022, but it still takes a significant number of reviewers and time to go through these applications. In fact, if reviewers take on average 1.89 hours per application (see line 206), the workload is reduced from 1,238 hours to 909 hours. This is an impressive reduction, but still a lot of work for the reviewers, and I wonder what can be done to improve the peer review process.

6) Finally, I miss numbers and scales in the graphs. It is difficult to compare graphs with numbers in the main text and captions. I'm also interested in why the authors used a visual analog scale. What are the reasons for not using gradients such as strongly agree, agree, ..., neither agree nor disagree, ... strongly disagree?

(Remarks to the Author)

Overall, this manuscript is original, with sound methodology, and has provided empirical evidence for the results, and for the existing literature on the topic of funding lottery.

However, I believe this manuscript can be greatly improved in terms of clarity and discussion. Regarding clarity, first, what is 'greater belief in success' on page 4 line 82-83?

Did you try/consider surveying those who was not selected by lottery? I think you might have done for the first round (939 applicants surveyed), then please add the composition data (how many out of 939 got (un)selected) in the paragraph at the bottom of the page 7. Add also how many out of the 82 applicants surveyed in the second round became successful eventually. Maybe also test whether there is significant difference between successful and unsuccessful applicants survey results.

What does the 'expression of interest' include, only a cover letter or something like CV, research proposal, or a short description of potential impact? Did you survey how much time was spent on preparing such an expression of interest? If yes, please report it briefly.

For better clarify, please add some percentage data into the paragraph between page 7 and 8. For instance, 939 (16%) participated in the survey, 500 (8.6%) applicants were invited to submit full proposals... 152 project were selected for funding (30.4% success rate out of 500 applications). Limitation of the study was not mentioned in the manuscript and should be added at the end.

To improve the results interpretation and further discussion, I think some important issues that might impact the lottery decision making have been missed. First, what about the risk of missing excellent proposals in the first round of lottery? Not unexpected, the lottery would decrease time and resources spent on peer reviewing more proposals, but its biggest risk/disadvantage is to miss excellent proposals which should be considered in the decision-making process with references from the existing literature. This can be considered very unfair by some applicants, especially unselected ones, at least in my peer review research interviews.

Second, how applicants/reviewers/funders perceive the lottery are also dependent on some other factors such as the scale/pool of applications, researchers' dependency on the grant programme (or say, funding alternatives within the landscape), researchers' career stages, and so on. It would be better if you could address some other existing funding lottery models and discuss some possible reasons for your case funder to apply lottery earlier than peer review.

Third, if you calculate and emphasize the economic saving to apply lottery first model, you should also discuss the case of re-using rejected proposals for other programmes which compensate for the waste of time in preparing proposals unsuccessful in the first place. I understand this is challenging to consider in your calculations, but you should at least discuss such scenarios.

You may consider sharing the whole survey questions for better transparency and reproducibility. I wonder whether this was any narratives/descriptive answers, if yes, you may consider giving some quotes on typical responses in the manuscript, such as what are your unsatisfied issues with the lottery first model?

Version 1:

Reviewer comments:

Reviewer #1

(Remarks to the Author)

The authors have answered my questions. I have only minor comments.

The Health Research Council of New Zealand did not use lotteries as a tie-breaker at the end of the process (line 79), as there were no ties in this system because people were not scored. The conditional lottery system is quite different from the tie-breaker system, which is explained in this recent paper that examines the taxonomy of lottery systems: DOI = 10.1093/reseval/rvae025.

Line 189 "these costs amounted to an average of €4.19 million", averaged over what? I expected to read about total costs. Perhaps these are average totals? Averaged over the simulations?

Line 320, allocating more reviewers to each proposal could be a selling point to researchers. They may lose at random, but if they do get reviewed it will likely be a better assessment.

Line 619 "Both approaches thus result in a similar proportion of funded applications". Might be better worded as a similar number of winners?

Line 730, is "valence" the correct word?

The authors might want to prepare a separate non-technical and short summary document aimed at funding staff.

Reviewer #2

(Remarks to the Author)

The article has gained from the changes. Contratulation! On the one hand, the article now refers more closely to previous research on the acceptance of lotteries in combination with peer review procedures, which helps to make the characteristics of the approach presented more clearly apparent. I also welcome the more detailed explanations and additional information on the "Freiraum" funding program in the main text and in the appendix.

On the other hand, I am convinced by the explanation presented in the "Answer to Comment 6" for reviewer 2: "We also would like to highlight that one advantage of the lottery-first approach is not only the lower cost but also the controllability of costs through the number of lottery tickets ". I have now better understood that the comparative cost calculation for the overall process was done in 2024. The overall process includes writing, processing and evaluating the applications. However, my objection in Comment 6 related solely to the workload for reviewers. I continue to see only a slight reduction in the number of applications. The reviewers still have to assess around 500 applications. Without the lottery, there were 655 applications in 2022. Compared to 2024, this is less, but still an order of magnitude where I cannot follow your arguments when you write: "Second, fewer applications requiring review due to an initial lottery could allow for improvements in peer review processes compared to the traditional approach" (316-318) or even suggest in line 320 that "more reviewers could be assigned to each application". In my view, the workload for all reviewers is not substantially reduced. No time and capacity is freed up for the experts to devote to other tasks or to prepare even more reviews.

Nonetheless, the overall process will be easier to control for the funders. I therefore suggest that this aspect of better controllability be emphasized more clearly in the article. For me, this is the convincing benefit of the lottery-first approach.

Reviewer #3

(Remarks to the Author)

Thank you, all the authors, for carefully responding to every single comment I made earlier. I think the manuscript has been greatly improved especially the clarify and discussion depth. Just pay attention to the Figure C on page 12 where part of the title is hidden and you might shorten the original title a bit to fit the page. Except this minor correction, I am satisfied with the revised manuscript and approve it to be published in this reviewer's eyes. Congratulations.

REVIEWER COMMENTS

Reviewer #1 (Remarks to the Author):

This is a genuinely exciting paper that examines a big shake-up to research funding. I was aware of the same idea being currently be applied in Denmark (<https://villumfonden.dk/en/group/grantsubarea/villum-experiment>) so it was a pleasant surprise to see results earlier than I expected. I congratulate the funders for trying this innovation, as this is the only way to improve the system.

The results were very interesting with enormous reductions in costs, which is a potentially huge benefit as many funding systems have become over-competitive and inefficient.

The study design was straightforward and appropriate.

Comment 1:

A result that I think needs more discussion is the enormous increase in applications after the policy change. I think this should be mentioned in the abstract, as total application numbers are a key concern for funding staff. The numbers went from 655 submitted to 4,622 and then 5,792, which is an enormous increase. I appreciate that the lack of difference in quality scores is comforting, but there are other potential problems.

Answer to Comment 1:

Dear Adrian, Thank you very much for the appreciation of our work. The increase in applications should also be considered under the different funding models. Initially, the first-come, first-serve model per definition has had a limited number of slots. This approach was introduced to reduce the workload for the institutions and reviewers. However, it was significantly criticised in the German media, since many more applicants had prepared an application and wanted to apply but all slots had already been taken in a short period of time. There is no information on how many applicants had invested time in the application in vain, but it is reasonable to assume that the number exceeded the $N = 655$ by far. In addition, it has to be noted that applications that did not enter the peer review process (in the first-come, first served procedure) or applicants whose expressions of interest were not drawn one year could easily adapt these for the lottery of the subsequent year. This implies that the project pool gets larger over the years with little extra effort, as long as the applicants deem the projects as described in the expression of interest as sufficiently innovative, helpful, and up-to-date. We did not include an item asking about this reuse of expression of interest in the surveys so far but plan to do so in the future.

Thanks also for pointing us to the Villum Foundation, of which we were not aware. They introduced exactly the same idea of a lottery-first approach in their funding scheme. We have now added a sentence to make explicit that there are also other foundations that are already using different methods, including the lottery-first approach. It is good to know that there are more empirical investigations in progress with the same focus.

The new passage now reads as follows (ll. 111-113):

“Here, we present data on an implementation of the lottery-first approach followed by peer review to allocate funding at a large German foundation (notably, the Danish Villum Foundation currently follows a very similar approach).”

Comment 2:

One potential issue is “stalking horse” applications, where several applications are submitted to the lottery in the hope of getting at least one ticket. For example, a lab-head may tell all their postdocs to submit vaguely worded applications, then they take-over any winning ticket with their own idea. There are potential issues here with bullying and misrepresentation. I note that 19 and 39 proposals of the randomly selected 500 did not submit a proposal (line 477) and perhaps these were spare slots that were no longer needed when a lab won 2 or more tickets?

Thinking more positively, the large increase in applications could good news as more ideas are being gathered from the community. This may increase the diversity in ideas, which was a key point made in Shahar Avin’s PhD thesis (reference number 11 on page 14).

Answer to Comment 2:

This is a valid concern for the lottery-first approach in general. In the present implementation, the Stiftung Innovation in der Hochschullehre (StIL) first checks whether there are identical or similar expressions of interest, and second, whether the later full application (of those being drawn) matches the content of the expression of interest. Furthermore, the grant is personalized to the person responsible for the application so that, at least in theory, each applicant has ownership over their own application and the grant money. Of course, these checks have limitations and it is still possible to increase lottery chances for a single work group by submitting multiple expressions of interest for different projects.

The added passage now reads as follows (ll.: 544-550):

“Applicants first submitted a short expression of interest (1,500 characters) in which they outlined the scope of their project and provided personal details, including name, gender, academic titles, email address and phone number for correspondence, institution, personal role at the institution, and subject area (N = 4,918 in 2023 and N = 6,033 in 2024). The expressions of interest were checked by the StIL for formal eligibility criteria: Specifically, identical or very similar expressions of interest from the same work group were excluded from the lottery.”

The other passage reads as follows (ll.: 557-562):

“Over all three application periods, both expressions of interest (only in the years 2023 and 2024) and the full proposals were first formally evaluated (e.g., formal checks,

correspondence between expressions of interest and full proposals where applicable) and then subjected to peer review.”

To test whether the “stalking horse” issue is present in our data from 2024, we investigated those applicants who were selected by lot, but did not submit a full proposal (n=39). We checked whether there were simultaneous full proposals submitted from the same institution (e.g. same university, medical school, etc.), even though this is a very rough approach as huge universities with many institutes and departments were included. For n=12 cases (30.1%) there was no other submission from the same institution. For another n=12 cases (30.1%) there were 1-2 other submissions from within the same institution. And for n=15 cases (39.8%) there were more than 2 submissions from the same institution. Even if this cannot rule out the issue of “stalking horse” applications completely, considering the size of the institutions and eligible personnel we do not evaluate this issue as extremely critical. Also, as noted above, the StIL critically evaluates whether there are identical or very similar expressions of interest from the same work group as well as checks whether there is a correspondence between expressions of interest and full proposals. Only then, proposals will be subjected to peer review.

Minor comments

- *“exclusion” is too strong a word (in my opinion) when talking about the bias against minorities. There has certainly been discrimination, but exclusion implies that there were no winners from these groups.*

Answer: Thanks for this remark. We have changed it accordingly.

- *The sentence starting in line 178 came too early, as the difference in costs was only made clear in the next paragraph.*

Answer: Thank you for pointing that out. We deleted that sentence as it was also repeated in the next paragraph after the cost of the conventional approach

- *39 hours (on average) seems relatively low for a full application and based on the list of what applicants had to provide (lines 462 to 464). I note that the question was worded in terms of hours (line 553) and the results may have been different if the scale was in days.*

Answer: This is a valid and interesting idea, and we will consider rephrasing this question in future questionnaires.

- *The almost 50:50 split in preference is fascinating (line 229).*

Answer: Yes, this is true, and we were also astonished. However, when speaking about the 50:50 split with scientists/colleagues, we often received the feedback that it was

“just” 50% - we hope we were able to show clearly that this is a very strong indicator of openness to change in the community.

- *Line 300, when the application costs exceed the available budget, this is known as the Szilard point, see https://en.wikipedia.org/wiki/Szilard_point*

We have added a reference to the original paper. The new passage now reads as follows (ll. 65-67):

“In some cases, the grant allocation process even depletes more resources from the system than they bring in, also known as the Szilard-point (Szilard 1961; Gross and Bergstrom 2019).”

- *Lines 522, 564, 571, the survey response percentages can be rounded and do not need 2 decimal places.*
- *The low survey response rates should likely be mentioned in the limitations as well as the likely direction of any bias. The response rates are typical for current surveys of busy academics.*

Answer: Thanks for the comment.

We have added a limitation which now reads as follows (ll. 606-610):

“Overall, although the survey response rates were rather low, they can be considered typical for current surveys of busy academics. Notably, there could be a bias within the responders in the way that only highly motivated applicants participated in the survey and therefore may bias results. This could for instance partly explain the bimodal distribution of reviewer preferences.”

- *I would add the average grant size per successful application. This will be one of the first questions for any funder looking to implement a similar scheme.*

Answer: We have added an additional supplemental table S5 with an overview of the grant sizes and distributions in the different funding periods. The average grant size per successful applicant was €288k in 2023 and €338k in 2024.

Reviewer #2 (Remarks to the Author):

The article deals with an important and much sought-after topic. Within the community of research funders and science policy makers, there is a great deal of interest in simplified grant distribution, including the early use of lotteries. For this reason, the results of the lottery-first approach presented here are likely to meet with great interest.

Answer: Thank you very much for appreciating our work!

Comment 1:

Against this background, it seems to me particularly important that the article in its current form is rewritten to include further studies on lottery procedures in research funding. I think this is extremely important because the results presented need to be seen and discussed in the context of other findings. The article is written in a style typical of the journal Nature, but the results presented are not original to the same extent. This does not detract from the importance of the results presented, but does call for a stronger contextualization of the findings.

I will list a few points and recommend that they be taken into account in the revision:

1) In some cases, the authors cite literature but do not fully consider its findings. For example, they cite Liu, M. et al. (2020) in the journal Research Integrity and Peer Review as a source for the Health Research Council (HRC). However, Liu and colleagues do not just describe the use of a lottery in the HRC funding process. They report on how applicants view the use of their lottery and what they accept and prefer. This finding strengthens the case for a particular type of lottery. The current paper is therefore not the first to investigate the acceptance of lotteries by researchers. The results should therefore be seen in the context of other findings. In particular, the authors should consider two surveys of the German scientific community that report scepticism and substantial rejection of random grant allocation. For example, the DZHW survey of scientists in 2019 shows that 69 percent of participants reject a random distribution of funding if the proposals are formally correct (Ambrasat, J. and Heger, C. 2020 in the report DZHW Barometer für die Wissenschaft). This finding is replicated in another survey of scientists in 2021, which shows that 66 percent reject a purely random distribution of grant proposals (Philipps, A. 2022 in the journal Science and Public Policy).

On the other hand, respondents to the Liu et al. and Philipps surveys support procedures that combine lotteries and peer review. This is also consistent with the observation by Philipps (2021 in the journal Research Evaluation) that German scientists who deliberate about the random distribution of grant proposals emphasise the importance of scientific institutions such as peer review. Peer review may be problematic in some respects, but it is considered the gold standard in science for checking the originality and feasibility of proposed research. In my view, this seems relevant to understanding the responses to the lottery-first approach in the survey reported here. The lottery-first approach presented is a combination of lottery and peer review (see line 530-537), and my reading is that this aspect is important to the participants. They are open to the lottery, but at the same time a large proportion of them is still in favour

of conventional peer review and they may support the lottery-first approach only because it includes a peer review process in the second round.

Answer to Comment 1:

Thank you very much for your comment, which introduces a remark on the pure lottery approach that has been suggested in the literature and was intensely criticized. To better integrate this perspective and the prior investigations on this type of lottery, we have now added the following passage citing the relevant work:

The added passage now reads as follows (ll. 73-81):

“To address some of these issues, lotteries have been proposed as an alternative way of distributing funds (Avin 2015, 2018, 2019; Ioannidis 2011; Liu et al. 2020). While the random allocation of grants is seen as increasing the chances of marginalized scientists and unconventional approaches to receive funding, the scientific community has greeted this idea with scepticism expressing fears of a threat to science due to a lack of quality control, as the gold standard of evaluation, peer review, is absent in a pure lottery (Philipps 2021; Ambrasat and Heger 2020). Against the backdrop, a combination of grant lottery and peer review was advocated, i.e. lotteries were introduced as a tie-breaker at the end of the decision process in funding allocations (e.g., by the Swiss National Science Foundation, the Health Research Council of New Zealand or the VolkswagenStiftung in Germany (Chawla 2021; Nature editorial 2022; Heyard et al. 2022; Liu et al. 2020; Bieri et al. 2021; Fang and Casadevall 2016; Philipps 2022; Philipps 2021)).”

Comment 2:

2) In addition, the authors cite Gross and Bergstrom (2019) in the journal PLoS Biology, but do not consider similar research. There are other studies that show how efficient a lottery approach can be (including acknowledging that random distribution is less costly, increases diversity, and increases opportunities for minorities). I provide this list of research because the results of the present study support what has been shown previously (especially regarding less work and cost):

Avin, S. 2015 in Recent Developments in the Philosophy of Science.

Avin, S. 2018 in RT. A Journal on Research Policy and Evaluation.

Bieri, M., Roser, K., Heyard, R., et al. 2021 in BMJ Open.

Höylä, T., Bartneck, C., and Tiihonen, T. 2016 in Scientometrics.

Answer to Comment 2:

Thank you for your suggestions for further literature. We were happy to follow your guidance, and have included references to Avin (2015), Avin (2018), Bieri et al (2021) and Hölyä et al. (2016) in the manuscript.

Comment 3:

3) In light of other research on lotteries and research funding, the authors should also reflect more on the "Freiraum" funding line. This initiative is aimed at scientists and science managers. It is not about basic research but about innovative concepts for higher education, i.e. it is not about potentially risky research. Rather, the concepts proposed are potential solutions based on proven knowledge. This may have an impact on who submits to "Freiraum". The focus on one particular funding initiative also limits the results. This limitation should be stated in the paper.

Answer to Comment 3:

Thank you for this remark. As we have described in the method section, the "Freiraum" model explicitly targets projects that aim to improve the quality of teaching in German higher education. This presumably introduces different biases regarding the nature of the proposals and the applicant pool, which might limit the generalizability of the results. Cost estimates are potentially less biased, and might even show smaller differences compared to a comparison of lottery-first vs a traditional approach with longer and more detailed full applications. Attitudes towards the funding line could be more in favour of a lottery, although this is difficult to argue in absence of empirical data. However, it must be noted that entry lotteries might also not be a generalizable solution for all funding lines. In some cases, they might be very worth considering, and in others less so. We now discuss these limitations in more detail.

Comment 4:

4) The paper also gives no information about the amount of money an approved proposal might receive. I would argue that the acceptance of a random grant allocation depends on the size of the grant amount.

Answer to Comment 4:

We agree that this is an important piece of information that should have been included in the initial version of the manuscript. We have now added the maximum amount of 400.000 € and the project duration of 24 months when describing the Freiraum funding line. In addition, we have included a supplemental table (S5) with more details on the grant sizes in each funding year. Regarding the impact of the amount of money on the acceptance of a distribution procedure, we believe that any innovation (rightfully) will face greater scrutiny when the amount of grant money increases. That might also come with greater criticism and increased highlighting and discussion of the risks of the new proposed mechanism. Since so far there is basically no data on acceptance of a lottery-first approach (plus peer review) and little data on tie-breaker lotteries, we think this might be an interesting empirical question for future research.

The new passage now reads as follows (ll.126-129):

“During the most recent implementation in 2024 (overall budget €50 million for a maximum of 400.000 € per project over 24 months), we collected data from several

waves of surveys with both applicants and reviewers (Fig 1B, see <https://osf.io/4ufrb/> for the original materials).”

Comment 5:

5) Lines 326-327 state: *"Notably, this preference for reduced workload outweighs applicants' perceptions of lower chances of success, quality, or fairness associated with the lottery-first approach."* What is meant by "outweighs"? Reduced applicant workload is a positive of lotteries, but it is not comparable to chances of success, quality, or fairness. It is clear that a lottery reduces the workload (see for example Shaw, J. 2022 in the journal *Research Evaluation*), but it is less clear that it addresses other shortcomings of peer review. However, scientists seem to be concerned about the quality of proposals when proposals are selected randomly, and established scientists in particular are sceptical because a lottery reduces their chances of success compared to a conventional peer review process (see Philipps 2022). How do the results presented agree or disagree with previous results?

Answer to Comment 5:

By "outweigh" we mean that for the overall preference for a lottery-first or traditional approach, the lower workload has such a strong impact that it makes the lottery-first approach equal to the preference for the traditional approach in terms of fairness, quality and chances of success. We have clarified this in the revised version of the manuscript (ll. 340-343):

“Notably, this preference for reduced workload outweighs applicants' perceptions of lower chances of success, quality, or fairness associated with the lottery-first approach, resulting in a preference distribution in which neither the lottery-first approach nor the traditional peer-review-only process is clearly preferred.”

Importantly, we do not speculate whether the reduced workload compensates for the other aspects *per se*. Instead, we describe that participants indeed view this as the most positive aspect which seems to influence the overall preference rating.

Comment 6:

5) *I was confused reading that "fewer applications requiring review could allow for improvements in peer review processes." (line 304) Of course, reviewing 481 full proposals in 2023 is less than the 655 in 2022, but it still takes a significant number of reviewers and time to go through these applications. In fact, if reviewers take on average 1.89 hours per application (see line 206), the workload is reduced from 1,238 hours to 909 hours. This is an impressive reduction, but still a lot of work for the reviewers, and I wonder what can be done to improve the peer review process.*

Answer to Comment 6:

Yes, this is worth pointing out. There are multiple things to consider: First and most important, the statement cited (now line 316) does not compare the different years of the Freiraum process. Instead, we compare the lottery-first approach to a

traditional one-stage full proposal process, the costs of which we estimated based on the survey responses and applications for the lottery-first approach. This is the data that is presented in Fig. 2. We rephrased the sentence to (ll. 316-318):

“Second, fewer applications requiring review due to an initial lottery could allow for improvements in peer review processes compared to the traditional approach”.

It is important to bear in mind that the number of applications in the traditional approach in 2022 would very likely have been significantly higher than 655 as the number was limited by the foundation as their first approach to reduce the total workload in the system. We also would like to highlight that one advantage of the lottery-first approach is not only the lower cost but also the controllability of costs through the number of lottery tickets.

Comment 7:

6) Finally, I miss numbers and scales in the graphs. It is difficult to compare graphs with numbers in the main text and captions. I'm also interested in why the authors used a visual analog scale. What are the reasons for not using gradients such as strongly agree, agree, ..., neither agree nor disagree, ... strongly disagree?

Answer to Comment 7:

Thank you for this remark. We have added numbers and scales in Fig. 3 matching the numerical space of the visual analogue in a style comparable to Fig. 2. We chose a visual analogue scale so that the response values for the individual items are on an interval scale compared to an ordinal scale produced by Likert and related scales. It also allows more fine-grained responses, leading to fewer errors.

Reviewer #3 (Remarks to the Author):

Overall, this manuscript is original, with sound methodology, and has provided empirical evidence for the results, and for the existing literature on the topic of funding lottery.

Comment 1:

However, I believe this manuscript can be greatly improved in terms of clarity and discussion. Regarding clarity, first, what is 'greater belief in success' on page 4 line 82-83?

Answer to Comment 1:

In our remark on “greater belief in success” we referred to the DFG Report on “Gender Effects in Research Funding” that is stating the following:

“Differences in personal and social responsibilities of male and female academics (marital status, children, domestic and elderly care, etc.) and in other individual factors (women's attitudes, motivation, self-confidence, fear of being perceived as highly assertive and confrontational, etc.) can significantly influence grant application behaviour and success, but have a lesser effect on their academic productivity.” (Ranga, Gupta, and Etzkowitz 2012; Sonnert and Holton 2006)

We summarized “other individual factors (women's attitudes, motivation, self-confidence, fear of being perceived as highly assertive and confrontational, etc.)” to “greater belief in success” as a standard psychological variable.

Now, in the revised version, we speak of “greater confidence” instead and hope to clarify things here (please refer to ll. 88-90).

“For instance, some applicants or groups of potential applicants may be more likely to submit proposals because they have more resources, better support structures, or more confidence than others (Ranga, Gupta, and Etzkowitz 2012; Sonnert and Holton 2006).”

Comment 2:

Did you try/consider surveying those who was not selected by lottery? I think you might have done the first round (939 applicants surveyed), then please add the composition data (how many out of 939 got (un)selected) in the paragraph at the bottom of the page 7. Add also how many out of the 82 applicants surveyed in the second round became successful eventually. Maybe also test whether there is significant difference between successful and unsuccessful applicants survey results.

Answer to Comment 2:

This is a suggestion that we would love to follow since we agree that the information would be very valuable. However, applicants were invited to the

survey separately from the foundation's interface for the application, and the responses did not allow cross-referencing to the actual applications for data protection reasons. Thus, we do not have the means to find out how many of the respondents of the first survey got a winning ticket, and how many of the second survey got accepted. We want to highlight, however, that the first survey took place before the lottery decision, so the results are independent of whether or not applicants got selected.

Comment 3:

What does the 'expression of interest' include, only a cover letter or something like CV, research proposal, or a short description of potential impact? Did you survey how much time was spent on preparing such an expression of interest? If yes, please report it briefly.

Answer to Comment 3:

We fully concur that clarifying what expressions of interest entail is an important element of contextualizing the findings. Although we had already done so in the methods section ("short expression of interest (1,500 characters) in which they outlined the scope of their project and provided personal details", l. 436), we have now added a more prominent explanation to the main text to aid the reader. The changed paragraph now reads as follows (ll. 544-548):

"Applicants first submitted a short expression of interest (1,500 characters) in which they outlined the scope of their project and provided personal details, including name, gender, academic titles, email address and phone number for correspondence, institution, personal role at the institution, and subject area (N = 4,918 in 2023 and N = 6,033 in 2024)."

In addition, we agree that displaying the amount of time spent on the expressions of interest is an important variable in this context. We have included this information in Figure 2A (second panel), showing the full distribution of applicants' and co-authors' reported time spent on the expression of interest. For convenience, we include the relevant part of this figure again below:

Fig. 2 Working hours and estimated costs for the lottery-first approach versus a conventional approach.

(A) Estimated working hours invested by various stakeholders during the application process for the Freiraum funding line, as implemented by the Stiftung Innovation in der Hochschullehre in 2024. Administrative working hours are divided into fixed costs for managing processes and variable costs that scale with the number of submissions at each stage. Applicants' self-reported working hours are derived from survey responses. For the initial expression of interest, applicants estimated an average preparation time of 6.1 hours (± 4.8 hours, standard deviation), with collaborators contributing an additional 4.7 hours (± 5.3 hours). For the full proposal stage, applicants reported an average preparation time of 39 hours (± 20 hours), while collaborators invested 33 hours (± 27 hours). Reviewers reported an average of 1.89 hours (± 0.97 hours) to evaluate a single full proposal.

Comment 4:

For better clarify, please add some percentage data into the paragraph between page 7 and 8. For instance, 939 (16%) participated in the survey, 500 (8.6%) applicants were invited to submit full proposals...152 project were selected for funding (30.4% success rate out of 500 applications).

Limitation of the study was not mentioned in the manuscript and should be added at the end.

Answer to Comment 4:

We agree that the reported results would be easier to follow if we included proportional information. We have added the information as requested in the caption of Fig. 1.

In addition, in line with your suggestion to discuss limitations regarding the response rate and to comply with a suggestion raised by reviewer 1, we have added a paragraph that now reads (lines: 606-610).

“Overall, although the survey response rates were rather low, they can be considered typical for current surveys of busy academics. Notably, there could be a bias within the responders in the way that only highly motivated applicants participated in the survey and therefore may bias results. This could for instance partly explain the bimodal distribution of reviewer preferences.”

We also discuss limitations regarding the potential loss of high-quality submissions in the discussion now (see ll. 344-359; see also your next comment):

“One argument that is often brought forward against the lottery-first approach is a presumed lower quality of funded proposals. There are several reasons against this assumption: First, reviewers’ quality assessments indicated that proposals submitted under the lottery-first approach were generally as strong as expected, allaying concerns that this method might significantly compromise the quality of proposals or subsequent research. Second, it is true that in a lottery-first approach, the quality of an application does not affect its probability of advancing. However, the extent to which this leads to a reduction in the quality of funded applications compared to the traditional approach depends on the accuracy of the peer review in both approaches, since traditional approaches are also not error-free. Given biases and unsystematic errors in peer review, the often very short initial applications in traditional approaches, and higher entry biases^{51,57,58}, it is not certain but rather an empirical question whether the lottery-first approach on average favours lower-quality proposals than a non-ideal traditional one. Probabilities of advancing are very easy to calculate in lottery-approaches, but very difficult to estimate for peer review. For this reason, in a previous publication we provided an interactive tool to compare different approaches in terms of quality, cost and fairness^{9,59} and called for more empirical research on grant quality.”

Comment 5:

To improve the results interpretation and further discussion, I think some important issues that might impact the lottery decision making have been missed. First, what about the risk of missing excellent proposals in the first round of lottery? Not unexpected, the lottery would decrease time and resources spent on peer reviewing more proposals, but its biggest risk/disadvantage is to miss excellent proposals which should be considered in the decision-making process with references from the existing literature. This can be considered very unfair by some applicants, especially unselected ones, at least in my peer review research interviews.

Answer to Comment 5:

This is indeed a counter-argument that is often put forward against the lottery procedure, and we agree that non-selected applicants often point out this perceived unfairness, as indicated by our results (ll. 242-243):

“However, the conventional approach was preferred for perceived chances of success, fairness, and the expected quality of funded applications.”

This is also evident in the open format responses we have now added in an extra section labelled “Exemplary Responses to Open Response Fields” (see ll., 727-797). Regarding missing excellent proposals, it is true that a lottery will not select the best proposals in general, but rather that each entry will have the same probability of being selected regardless of quality. Therefore, the number of excellent proposals that are missed (or selected) in the lottery depends on the number of lottery tickets. However, avoiding built-in biases, the manuscript focuses on the puristic case of equal chances for all proposals. We think it is important to not compare the lottery-first procedure to the best possible procedure but to the traditional and currently dominant funding allocation model: peer-review only. Our analysis shows that reviewers found the quality of the proposals to be broadly comparable in both models, which we have now clarified further in the manuscript.

Lines 245-249:

“Importantly, the assessed quality of the proposals was in line with the reviewers' expectations, with no indication of unusually low-quality submissions on average (Figure 3D). Concerns that the quality of proposals submitted to peer review might suffer from the implementation of a lottery-first procedure were therefore mitigated.”

Lines 340-360:

“Notably, this preference for reduced workload outweighs applicants' perceptions of lower chances of success, quality, or fairness associated with the lottery-first approach, resulting in a preference distribution in which neither the lottery-first approach nor the traditional peer-review-only process is clearly preferred. One argument that is often brought forward against the lottery-first approach is a presumed lower quality of funded proposals. There are several reasons against this assumption: First, reviewers' quality assessments indicated that proposals submitted under the lottery-first approach were generally as strong as expected, allaying concerns that this method might significantly compromise the quality of proposals or subsequent research. Second, it is true that in a lottery-first approach, the quality of an application does not affect its probability of advancing. However, the extent to which this leads to a reduction in the quality of funded applications compared to the traditional approach depends on the accuracy of the peer review in both approaches, since traditional approaches are also not error-free. Given biases and unsystematic errors in peer review, the often very short initial

applications in traditional approaches, and higher entry biases (Graves, Barnett, and Clarke 2011; Jerrim and Vries 2023; Pier et al. 2018), it is not certain but rather an empirical question whether the lottery-first approach on average favours lower-quality proposals than a non-ideal traditional one. Probabilities of advancing are very easy to calculate in lottery-approaches, but very difficult to estimate for peer review. For this reason, in a previous publication we provided an interactive tool to compare different approaches in terms of quality, cost and fairness (Luebber et al. 2023; Hölyä, Bartneck, and Tiihonen 2016) and called for more empirical research on grant quality.”

Given that initial applications for lottery-first approaches are often short (to reduce costs), and reviews can be biased against groundbreaking ideas, the question about the proportion of misses of excellent proposals is an empirical one: What degree of bias in the traditional model would lead to outcomes similar to the lottery-first approach? In the absence of informative empirical data, we have provided a simulation addressing this question (see (Luebber et al. 2023) for further details regarding the Shiny app).

Comment 6:

Second, how applicants/reviewers/funders perceive the lottery are also dependent on some other factors such as the scale/pool of applications, researchers’ dependency on the grant programme (or say, funding alternatives within the landscape), researchers’ career stages, and so on. It would be better if you could address some other existing funding lottery models and discuss some possible reasons for your case funder to apply lottery earlier than peer review.

Answer to Comment 6:

Thanks for alerting us to the opportunity to further contextualize our findings. We fully concur, and have altered the manuscript accordingly.

We now explicate why the foundation chose to apply a lottery-first procedure (see ll. 535-540):

“From the beginning, procedures for submitting to the funding line were designed to save costs in the allocation process.”

We now also point out more explicitly that the foundation chose to try two different allocation procedures to save costs (see ll. 536-540):

“Two different allocation mechanisms were used: First, when the funding line was established in 2022, it followed a first-come, first served procedure for a total of 600 proposals (including, among others, a description of the problem or state of the art, an outline of the project plan including objectives and work plan, description

of the project's contribution to developing the state of the art, and financial details).”

In addition, we now discuss the prominent role of the funding line (see ll. 530-534):

“Overall, the funding line stands out in the German context due to its broad eligibility criteria and large funding volume dedicated to higher education. Comparable funding programs are available from the StIL itself (€330 million dedicated to digitalization in higher education), while other funding for innovation in higher education is usually allocated via smaller-scale funding lines.”.

Comment 7:

Third, if you calculate and emphasize the economic saving to apply lottery first model, you should also discuss the case of re-using rejected proposals for other programmes which compensate for the waste of time in preparing proposals unsuccessful in the first place. I understand this is challenging to consider in your calculations, but you should at least discuss such scenarios.

Answer to Comment 7:

This is a very valid point. In general, this is an aspect of grant cost estimation that is difficult to model and poses a problem for all scenarios, not just the lottery-first approach, as in principle any unsuccessful grant application can be reused to some degree for the same or other calls. The degree to which this impacts the cost comparison is thus hard to tell. However, re-using proposals is possible regardless of whether a lottery-first or peer-review-only allocation mechanism is employed and therefore, in expectation, affects our cost estimations for both approaches equally. We now note in the methods section that both estimates could be biased. The relevant passage reads (see ll. 709-710):

“Note that because cost estimates are based on self-reported working hours, they may differ from the true value due to systematic reporting biases.”

We nevertheless now discuss the possibility of reusing materials, as well as other sources of potential between-participant sources of variation (see ll. 711-713):

“In addition, we have not modelled potential sources of variation between participants, such as whether they could reuse materials from an earlier unsuccessful proposal, their available time budgets, or career status.”

For our future investigations, we will add an item asking about the reuse of materials.

Comment 8:

You may consider sharing the whole survey questions for better transparency and reproducibility. I wonder whether this was any narratives/descriptive answers, if yes, you may consider giving some quotes on typical responses in the manuscript, such as what are your unsatisfied issues with the lottery first model?

Answer to Comment 8:

Thanks for pointing out that we could further increase the transparency of our reporting by sharing the original materials, in addition to the data we have already provided. We have made the surveys for the three data collection phases reported here available in the supplemental material. In the manuscript, we point readers to this resource, writing (see ll. 127-129):

“[...] we collected data from several waves of surveys with both applicants and reviewers (Fig 1B, see <https://osf.io/4ufrb/> for the original materials).”

Further, we fully concur that readers may find it interesting to read some of the responses given to the open feedback items. We have now included exemplary responses in the supplement. For convenience, please find below two of these exemplary responses, but see “Exemplary Responses to Open Response Fields” (ll. 727-797) for more examples:

“I hope that luck will help me get out of the lottery drum and into the process. Thank you for this great opportunity from the foundation! (English translation)”

“I think an application process with an expression of interest and an estimated draw rate of 2.5% (I'm assuming 20,000 interested parties) is completely naive and academically ridiculous in the age of ChatGPT. In short, a debacle for the foundation. (English translation)”

References

- Ambrosat, J., and C. Heger. 2020. “Barometer Für Die Wissenschaft: Ergebnisse Der Wissenschaftsbefragung 2019/20.” Berlin: Deutsches Zentrum für Hochschul- und Wissenschaftsforschung (DZHW).
- Avin, Shahar. 2015. “Funding Science by Lottery.” In *Recent Developments in the Philosophy of Science: EPSA13 Helsinki*, 111–26. Cham: Springer International Publishing.
- . 2018. “Policy Considerations for Random Allocation of Research Funds.” *RT. A Journal on Research Policy and Evaluation* 6 (1). <https://doi.org/10.13130/2282-5398/8626>.
- . 2019. “Mavericks and Lotteries.” *Studies in History and Philosophy of Science* 76 (August): 13–23.
- Bieri, Marco, Katharina Roser, Rachel Heyard, and Matthias Egger. 2021. “Face-to-Face Panel Meetings versus Remote Evaluation of Fellowship Applications: Simulation Study at the Swiss National Science Foundation.” *BMJ Open* 11 (5): e047386.

- Chawla, Singh. 2021. "Swiss Funder Draws Lots to Make Grant Decisions." Nature Publishing Group UK. May 6, 2021. <https://doi.org/10.1038/d41586-021-01232-3>.
- Fang, Ferric C., and Arturo Casadevall. 2016. "Research Funding: The Case for a Modified Lottery." *MBio* 7 (2): e00422-16.
- Graves, Nicholas, Adrian G. Barnett, and Philip Clarke. 2011. "Funding Grant Proposals for Scientific Research: Retrospective Analysis of Scores by Members of Grant Review Panel." *BMJ* 343 (September): d4797.
- Gross, Kevin, and Carl T. Bergstrom. 2019. "Contest Models Highlight Inherent Inefficiencies of Scientific Funding Competitions." *PLoS Biology* 17 (1): e3000065.
- Heyard, Rachel, Manuela Ott, Georgia Salanti, and Matthias Egger. 2022. "Rethinking the Funding Line at the Swiss National Science Foundation: Bayesian Ranking and Lottery." *Statistics and Public Policy* 9 (1): 110–21.
- Hölyä, T., C. Bartneck, and T. Tiihonen. 2016. "The Consequences of Competition: Simulating The Effects Of Research Grant Allocation Strategies." *Scientometrics* 108 (1): 263–88.
- Ioannidis, John P. A. 2011. "More Time for Research: Fund People Not Projects: More Time for Research." *Nature* 477 (7366): 529–31.
- Jerrim, John, and Robert Vries. 2023. "Are Peer Reviews of Grant Proposals Reliable? An Analysis of Economic and Social Research Council (ESRC) Funding Applications." *The Social Science Journal* 60 (1): 91–109.
- Liu, Mengyao, Vernon Choy, Philip Clarke, Adrian Barnett, Tony Blakely, and Lucy Pomeroy. 2020. "The Acceptability of Using a Lottery to Allocate Research Funding: A Survey of Applicants." *Research Integrity and Peer Review* 5 (1): 3.
- Luebber, Finn, Sören Krach, Marina Martinez Mateo, Frieder M. Paulus, Lena Rademacher, Rima-Maria Rahal, and Jule Specht. 2023. "Rethink Funding by Putting the Lottery First." *Nature Human Behaviour* 7 (7): 1031–33.
- Nature editorial. 2022. "The Case for Lotteries as a Tiebreaker of Quality in Research Funding." *Nature* 609 (653): 653–653.
- Philipps, A. 2021. "Science Rules! A Qualitative Study of Scientists' approaches to Grant Lottery." *Research Evaluation* 30 (1): 102–11.
- Philipps, Axel. 2022. "Research Funding Randomly Allocated? A Survey of Scientists' Views on Peer Review and Lottery." *Science & Public Policy* 49 (3): 365–77.
- Pier, Elizabeth L., Markus Brauer, Amarette Filut, Anna Kaatz, Joshua Raclaw, Mitchell J. Nathan, Cecilia E. Ford, and Molly Carnes. 2018. "Low Agreement among Reviewers Evaluating the Same NIH Grant Applications." *Proceedings of the National Academy of Sciences of the United States of America* 115 (12): 2952–57.
- Ranga, Marina, Namrata Gupta, and Henry Etzkowitz. 2012. "Gender Effects in Research Funding A Review of the Scientific Discussion on the Gender-Specific Aspects of the Evaluation of Funding Proposals and the Awarding of Funding." Deutsche Forschungsgemeinschaft (DFG). <https://www.dfg.de/resource/blob/170570/e48fab44b49274b83e2b5aeb382145d0/studie-gender-effects-data.pdf>.
- Sonnert, Gerhard, and Gerald Holton. 2006. *Who Succeeds in Science? The Gender Dimension*. New Brunswick: Rutgers University Press.
- Szilard, Leo. 1961. *The Voice Of the Dolphins and Other Stories*. New York: Simon and Schuster.

E1

Thank you for submitting your manuscript "Reducing gender bias and economic cost of science funding by putting a lottery first" to Nature Communications. I am delighted to say that we are happy, in principle, to publish it under an open access license. Please accept our apologies for the long delay in getting back to you.

Response

Thank you for this positive feedback! We include our response to the comments below.

R1.1

The authors have answered my questions. I have only minor comments.

Response

We're happy to see that our responses to your comments addressed your concerns!

R1.2

The Health Research Council of New Zealand did not use lotteries as a tie-breaker at the end of the process (line 79), as there were no ties in this system because people were not scored. The conditional lottery system is quite different from the tie-breaker system, which is explained in this recent paper that examines the taxonomy of lottery systems: DOI = 10.1093/reseval/rvae025.

Response

Thank you for the clarification and the additional suitable reference. We modified the respective part, saying:

Against the backdrop, a combination of grant lottery and peer review was advocated, i.e. lotteries were introduced as a tie-breaker at the end of the decision process in funding allocations (e.g., by the Swiss National Science Foundation, or the VolkswagenStiftung in Germany) ^{7,23,24,26-30} or to select applications after an initial screening (Explorer Grant from the Health Research Council of New Zealand) ³¹.

R1.3

Line 189 “these costs amounted to an average of €4.19 million”, averaged over what? I expected to read about total costs. Perhaps these are average totals? Averaged over the simulations?

Response

Yes, these costs are averaged over all simulation runs. As there are random elements involved, predominantly assigning work hours to applications, not all runs have exactly the same costs. However, for the lottery-first procedure the spread is very small as can be seen in the error bars in figure 2D which represent minimum and maximum values. For the counterfactual traditional approach, the spread is larger as the number of applications is also a random draw (see R2.2 below). We added this detail in the main text:

“For 2024, these costs amounted to an average of €4.19 million **across all simulation runs**, [...]”

R1.4

Line 320, allocating more reviewers to each proposal could be a selling point to researchers. They may lose at random, but if they do get reviewed it will likely be a better assessment.

Response

We also believe that to be a very important aspect. To even more highlight this potential, we rephrased that paragraph to:

“Resources could be redirected toward **reviewer training**⁵³, implementing debiasing interventions^{54,55} and better familiarizing reviewers with unconventional proposals. Additionally, more reviewers could be assigned to each application, increasing the reliability of evaluations and better reconciling divergent opinions, particularly for innovative or groundbreaking ideas⁵⁶. **Since an acceptable reliability requires many more reviewers than is standard practice, with some estimates going as high as ten⁵⁷ or twelve⁵⁸, having fewer applications to review could enable a much higher-quality selection.**”

R1.5

Line 619 “Both approaches thus result in a similar proportion of funded applications”.
Might be better worded as a similar number of winners?

Response

We changed that part according to your suggestion.

R1.6

Line 730, is “valence” the correct word?

Response

We changed that sentence to:

“Below, we provide exemplary responses that cover **the broad spectrum of opinions**, in the original German and translated to English.”

R1.7

The authors might want to prepare a separate non-technical and short summary document aimed at funding staff.

Response

Thank you for this remark! We drafted such a summary and attempt to publish it alongside the publication.

R2.1

The article has gained from the changes. Congratulation! On the one hand, the article now refers more closely to previous research on the acceptance of lotteries in combination with peer review procedures, which helps to make the characteristics of the approach presented more clearly apparent. I also welcome the more detailed explanations and additional information on the “Freiraum” funding program in the main text and in the appendix.

On the other hand, I am convinced by the explanation presented in the “Answer to Comment 6” for reviewer 2: “We also would like to highlight that one advantage of the lottery-first approach is not only the lower cost but also the controllability of costs through the number of lottery tickets”. I have now better understood that the comparative cost calculation for the overall process was done in 2024. The overall process includes writing, processing and evaluating the applications.

Response

Thank you! We are glad that you are satisfied with the changes we made to the manuscript.

We now highlight the higher controllability both in the abstract:

“Thus, the lottery-first approach is a promising addition to allocation procedures, reducing some of its downsides, while offering higher controllability of costs for funding agencies by controlling the number of winning tickets.”

And in the conclusion:

“[...] remaining broadly acceptable to both applicants and reviewers. It also offers more controllability for funders since they can determine the number of reviews in advance by deciding about the number of winning tickets. By rethinking how funding is distributed [...]”

R2.2

However, my objection in Comment 6 related solely to the workload for reviewers. I continue to see only a slight reduction in the number of applications. The reviewers still have to assess around 500 applications. Without the lottery, there were 655 applications in 2022. Compared to 2024, this is less, but still an order of magnitude where I cannot follow your arguments when you write: “Second, fewer applications requiring review due to an initial lottery could allow for improvements in peer review processes compared to the traditional approach” (316-318) or even suggest in line 320 that “more reviewers could be assigned to each application”. In my view, the workload for all reviewers is not substantially reduced. No time and capacity is freed up for the experts to devote to other tasks or to prepare even more reviews.

Nonetheless, the overall process will be easier to control for the funders. I therefore suggest that this aspect of better controllability be emphasized more clearly in the article. For me, this is the convincing benefit of the lottery-first approach.

Response

Thank you for giving us another shot at clarifying this. It is important to note that we never intended to compare the reviewer workload between the years with a lottery-first approach, and the year with the first-come, first-served procedure, as both were designed with the aim to reduce reviewer workload. However, we were interested in the

comparison to a traditional peer review-only approach. We would thus agree with your statement that the reduction from 2022 to the other years is not very meaningful. What we want to do instead is to compare the reviewer workload in the lottery-first approach to the traditional approach where applicants submit the full application in the initial step. Since the foundation did not actually implement this procedure in any year, we simulated such a counterfactual scenario. For this to work, we needed an estimate of the number of full applications in such a counterfactual scenario - after all it is unreasonable to assume that this number would be equal to the number of expressions of interest, since the workload for a full application is much higher. Thus, we were using the survey item about the self-estimated likelihood of submitting a full application instead of an expression of interest, had the foundation implemented this traditional approach. This probability was then used to draw the actual sample of applicants in this scenario in each simulation run and their workloads, followed by drawing workloads of peer-reviewers. For convenience, we copy the relevant paragraph from the methods section here:

“The calculation of the potential costs of a conventional peer-review approach, was conducted as follows: First, it was assumed that all initial 6,033 *Freiraum* applicants are potential applicants in a conventional peer review approach but that not all would apply due to higher initial resource investment. Thus, whether each of the 6,033 applicants would have submitted a full proposal was determined by drawing a respondent’s certainty to submit a full proposal as rated on the visual analogue scale in survey 1 (0% - 100%) as input for the probability in a Bernoulli distribution. Second, if this draw resulted in a decision to submit a full proposal, the applicant’s working hours and the collaborators’ corresponding working hours were drawn from the respective responses in survey 2. If this draw resulted in a decision not to submit, no working hours were counted for this applicant. Third, the working hours for reviewing the full proposal were determined by two draws from the distribution of the invested work hours of peers in survey 3, excluding student reviewers since they received fixed compensation (see below). ”

R3.1

Thank you, all the authors, for carefully responding to every single comment I made earlier. I think the manuscript has been greatly improved especially the clarify and discussion depth.

Response

Thank you for your positive feedback!

R3.2

Just pay attention to the Figure C on page 12 where part of the title is hidden and you might shorten the original title a bit to fit the page. Except this minor correction, I am satisfied with the revised manuscript and approve it to be published in this reviewer's eyes. Congratulations.

Response

We believe that to be a misunderstanding, as the dot-dot-dot is supposed to trail into the x-axis labels and does not indicate missing text. However, we changed the title to avoid confusion:

“Preferences by aspect of the funding process”

Proposal for a short summary for interested funding agencies (as recommended by Reviewer 1):

Current research funding systems are under increasing strain. Application numbers continue to rise, but success rates remain low. The result: an expensive, time-consuming process for both researchers and funding agencies, with growing evidence of systematic bias that limits diversity of ideas in science.

We tested a simple but powerful change—a *lottery-first* approach. Instead of asking every applicant to submit a full proposal, a random draw determines which applicants advance to the detailed peer review stage. This allows funding agencies to control the number of full applications, cut costs on either side, and reduce bias.

Evidence from a large German funding program shows that this approach works:

- **Less bias** – Female applicants saw a 10% increase in submissions and a 23% increase in funded projects compared to the previous system.
- **Lower costs** – The lottery-first process reduced time to prepare and review proposals by 68% compared to standard single-stage peer review.
- **High acceptance** – Reviewers and applicants broadly supported the change; around half of applicants preferred it to the conventional process.

For policymakers and funding organizations, the implications are clear: adopting a lottery-first system offers a route to make funding allocation more efficient, more equitable, more sustainable and more controllable. Savings from this new approach can be reinvested into improving peer review, supporting researchers for the full proposals, but also reduce researcher's working time resulting in unsuccessful applications.